# Lamina-specific cortical dynamics in human visual and sensorimotor cortices

**James J Bonaiuto[1,2]\*, Sofie S Meyer[1,3,4], Simon Little[2], Holly Rossiter[5], Martina F Callaghan[1], Frederic Dick[6], Gareth R Barnes[1†], Sven Bestmann[1,2†]**

[1]Wellcome Centre for Human Neuroimaging, UCL Queen Square Institute of Neurology, University College London, London, United Kingdom; [2]Department for Movement and Clinical Neurosciences, UCL Queen Square Institute of Neurology, University College London, London, United Kingdom; [3]UCL Institute of Cognitive Neuroscience, University College London, London, United Kingdom; [4]UCL Queen Square Institute of Neurology, University College London, London, United Kingdom; [5]CUBRIC, School of Psychology, Cardiff University, Cardiff, United Kingdom; [6]Department of Psychological Sciences, Birkbeck College, University of London, London, United Kingdom

**Abstract** Distinct anatomical and spectral channels are thought to play specialized roles in the communication within cortical networks. While activity in the alpha and beta frequency range (7 – 40 Hz) is thought to predominantly originate from infragranular cortical layers conveying feedback-related information, activity in the gamma range (>40 Hz) dominates in supragranular layers communicating feedforward signals. We leveraged high precision MEG to test this proposal, directly and non-invasively, in human participants performing visually cued actions. We found that visual alpha mapped onto deep cortical laminae, whereas visual gamma predominantly occurred more superficially. This lamina-specificity was echoed in movement-related sensorimotor beta and gamma activity. These lamina-specific pre- and post- movement changes in sensorimotor beta and gamma activity suggest a more complex functional role than the proposed feedback and feedforward communication in sensory cortex. Distinct frequency channels thus operate in a lamina-specific manner across cortex, but may fulfill distinct functional roles in sensory and motor processes.

DOI: https://doi.org/10.7554/eLife.33977.001

\*For correspondence:
jbonaiuto@gmail.com

†These authors contributed equally to this work

**Competing interests:** The authors declare that no competing interests exist.

## Introduction

The cerebral cortex is hierarchically organized via feedback and feedforward connections that originate predominantly from deep and superficial layers, respectively (*Felleman and Van Essen, 1991*; *Barone et al., 2000*; *Markov et al., 2013*; *Markov et al., 2014a*; *Markov et al., 2014b*). Evidence from non-human animal models suggests that information along those pathways is carried via distinct frequency channels: lower frequency (<40 Hz) signals predominantly arise from deeper, infragranular layers, whereas higher frequency (>40 Hz) signals stem largely from more superficial, supragranular layers (*Roopun et al., 2006*; *Roopun et al., 2010*; *Bollimunta et al., 2008*; *Bollimunta et al., 2011*; *Sun and Dan, 2009*; *Maier et al., 2010*; *Buffalo et al., 2011*; *Spaak et al., 2012*; *Xing et al., 2012*; *Smith et al., 2013*; *van Kerkoerle et al., 2014*; *Bastos et al., 2015*; *Haegens et al., 2015*; *Sotero et al., 2015*). These data have inspired general theories of the functional organization of cortex which ascribe specific computational roles to these pathways and frequency channels (*Fries, 2005*; *Fries, 2015*; *Friston and Kiebel, 2009*; *Wang, 2010*; *Jensen and Mazaheri, 2010*; *Donner and Siegel, 2011*; *Arnal and Giraud, 2012*; *Bastos et al., 2012*; *Adams et al., 2013*; *Jensen et al., 2015*; *Stephan et al., 2017*). In these proposals, lower frequency

**eLife digest** As we interact with the world around us, signals flow from neuron to neuron and from one brain area to the next. When we look at an object, for example, signals pass along a pathway of areas in the outermost part of the brain, called the cortex. Each area along this visual pathway performs more complex processing than the one before it. But information also flows in the opposite direction along such cortical pathways. These feedback signals enable areas further along the pathway to influence the activity of those before them.

Studies in animals suggest that much like a highway, information is travelling in opposite directions within the cortex along different lanes. In mammals, these lanes consist of distinct layers of cells. In the visual cortex of monkeys, feedback signals travel via deeper layers of cortex, whereas feedforward signals travel via the upper layers. Brain activity in the upper layers also has a higher frequency than that in the lower layers.

But is this also the case in our own brains? Bonaiuto et al. used a technique called MEG to measure the frequency of brain activity within the upper and lower layers of cortex in healthy volunteers. The volunteers had to look at images on a screen and then respond by pressing a button. Bonaiuto et al. observed that activity in deeper layers of cortex occurred mostly at lower frequencies, while activity in upper layers mostly happened at higher frequencies. This pattern, which matches that seen in monkeys, was found in both visual cortex and in areas of cortex that help plan and execute movements. In visual cortex, the activity in the upper layers appeared to carry feedforward signals. But in movement-related areas, feedback and feedforward signals were less clearly related to cortical layers.

These findings lend support to current theories about how the cortex is organized. They also show that MEG can reveal rapidly changing brain activity at a high spatial resolution. The findings may also provide clues to the origins of brain disorders called oscillopathies. These involve changes in specific frequencies of brain activity, and include schizophrenia and epilepsy, among others.

DOI: https://doi.org/10.7554/eLife.33977.002

activity subserves feedback, top-down communication conveyed predominantly via infragranular layers, whereas high-frequency activity is predominantly carried via projections from supragranular layers and conveys feedforward, bottom-up information.

However, evidence for these proposals in humans is largely indirect, and focused on visual and auditory areas (*Koopmans et al., 2010*; *Olman et al., 2012*; *Fontolan et al., 2014*; *Kok et al., 2016*; *Michalareas et al., 2016*; *Scheeringa and Fries, 2017*). Whether it is indeed possible to attribute low and high frequency activity in humans to lamina-specific sources, throughout the cortical hierarchy, remains unclear. Here we leverage recent advances in high precision magnetoencephalography (*Troebinger et al., 2014b*; *Meyer et al., 2017a*) to address this issue directly and non-invasively across human visual and sensorimotor cortices.

MEG is a direct measure of neural activity (*Hämäläinen et al., 1993*; *Baillet, 2017*), with millisecond temporal precision that allows for delineation of brain activity across distinct frequency bands. Recently developed 3D printed head-cast technology gives us more stability in head positioning as well as highly precise models of the underlying cortical anatomy. Together, this allows recording of higher signal-to-noise ratio (SNR) MEG data than previously achievable (*Troebinger et al., 2014b*; *Meyer et al., 2017a*). Theoretical and simulation work shows that these gains allow, in principle, for distinguishing the MEG signal originating from either deep or superficial laminae (*Troebinger et al., 2014a*), in a time-resolved and spatially localized manner (*Bonaiuto et al., 2018*). Demonstrating such lamina-specificity non-invasively in healthy human participants would provide important physiological constraints to the development of theoretical accounts about the functional roles of different frequency channels, in particular with regards to the proposed mechanism of inter-regional communication in hierarchical cortical networks. Here, we employed this approach to acquire high SNR MEG data, and directly test for the proposed lamina-specificity of distinct frequency channels in human cortex.

## Results

### Behavioral responses vary with perceptual evidence and cue congruence

We investigated the laminar and spectral specificity of induced visual and sensorimotor activity during a visually cued action selection task. The task was designed to induce well-studied patterns of low- and high-frequency activity in visual (*Müller et al., 1996*; *Hari, 1997*; *Fries et al., 2001*; *Busch et al., 2004*; *Sauseng et al., 2005*; *Yamagishi et al., 2005*; *Hoogenboom et al., 2006*; *Thut et al., 2006*; *Muthukumaraswamy and Singh, 2013*; *Mazaheri et al., 2014*) and sensorimotor cortices (*Pfurtscheller et al., 1996*; *Pfurtscheller and Neuper, 1997*; *Crone et al., 1998*; *Cheyne et al., 2008*; *Donner et al., 2009*; *Huo et al., 2010*; *Gaetz et al., 2011*; *Haegens et al., 2011*; *de Lange et al., 2013*; *Tan et al., 2016*; *Tan et al., 2014*; *Torrecillos et al., 2015*). Participants first viewed a random dot kinematogram (RDK) with coherent motion to the left or the right, which in most trials (70%) was congruent to the direction of the following instruction cue indicating the required motor response (an arrow pointing left equated to an instruction to press the left button, and vice versa; *Figure 1A*). Participants could therefore accumulate the sensory evidence from the RDK to anticipate the likely required response in advance of the instruction cue. However, in incongruent trials, the instruction cue pointed in the opposite direction from the direction of coherent motion of the RDK, and so the opposite response from the expected one was required. The strength of the motion coherence varied between trials, thereby influencing the predictability of the instructed response (*Figure 1B*; *Donner et al., 2009*; *de Lange et al., 2013*).

As expected, particpants responded more accurately and quickly during congruent trials, with additional gains in respond speed when the RDK motion coherence was strongest. By contrast, responses were generally slower and participants made more mistakes during incongruent trials (*Figure 1C,D*). This was demonstrated by a significant interaction between congruence and coherence for accuracy ($\chi^2(2)$ = 363.21, $p<0.001$), and RT ($F(2,16187)$ = 25.83, $p<0.001$). Pairwise comparisons (Tukey corrected) showed that accuracy was higher and RTs were faster during congruent trials than incongruent trials at low (accuracy: $Z$ = 7.83, $p<0.001$; RT: $t(16181.94)$ = $-8.25$, $p<0.0001$), medium (accuracy: $Z$ = 23.71, $p<0.001$; RT: $t(16181.94)$ = $-13.94$, $p<0.001$) and high coherence levels (accuracy: $Z$ = 29.96, $p<0.001$; RT: $t(16181.94)$ = $-18.39$, $p<0.001$). Participants were thus faster and more accurate when the cued action matched the action they had prepared (congruent trials), and slower and less accurate when these actions were incongruent.

### High SNR MEG recordings using individualized head-casts

Participant-specific head-casts minimize both within-session movement and co-registration error (*Troebinger et al., 2014b*; *Meyer et al., 2017a*). This ensures that when MEG data are recorded over separate days, the brain remains in the same location with respect to the MEG sensors. In all participants, within-session movement was <0.2 mm in the x and y dimensions, and <1.5 mm in the z dimension, while co-registration error was <1.5 mm in any dimension (estimated by calculating the within-participant standard deviation of the absolute coil locations across recording blocks; *Figure 2—figure supplement 1*). To assess the between-session reproducibility of our data, we examined topographic maps, event-related fields (ERFs), and time-frequency (TF) decompositions for the different task epochs. These data were analyzed in three ways: aligned to the onset of the RDK (*Figure 2A*), instruction cue (*Figure 2B*), or button response (*Figure 2C*). Topographic maps and event-related fields from individual MEG sensors and time-frequency spectra from sensor clusters were indeed highly reproducible across different days of recording within an individual. For the participant shown in *Figure 2*, the intra-class correlation coefficient (ICC), a measure of test-retest reliability, was greater than 0.9 for all task epochs, and the three measures used to assess reproducibility (topographic map, RDK, mean within-session ICC = 0.95, between-session ICC = 0.94; topographic map, instruction cue, mean within-session ICC = 0.94, between-session ICC = 0.97; topographic map, button response, mean within-session ICC = 0.97, between-session ICC = 0.99; ERF, RDK, mean within-session ICC = 0.94, between-session ICC = 0.97; ERF, instruction cue, mean within-session ICC = 0.96, between-session ICC = 0.96; ERF, button response, mean within-session ICC = 0.96, between-session ICC = 0.98; TF, RDK, mean within-session ICC = 0.97, between-session ICC = 0.97; TF, instruction cue, mean within-session ICC = 0.97, between-session

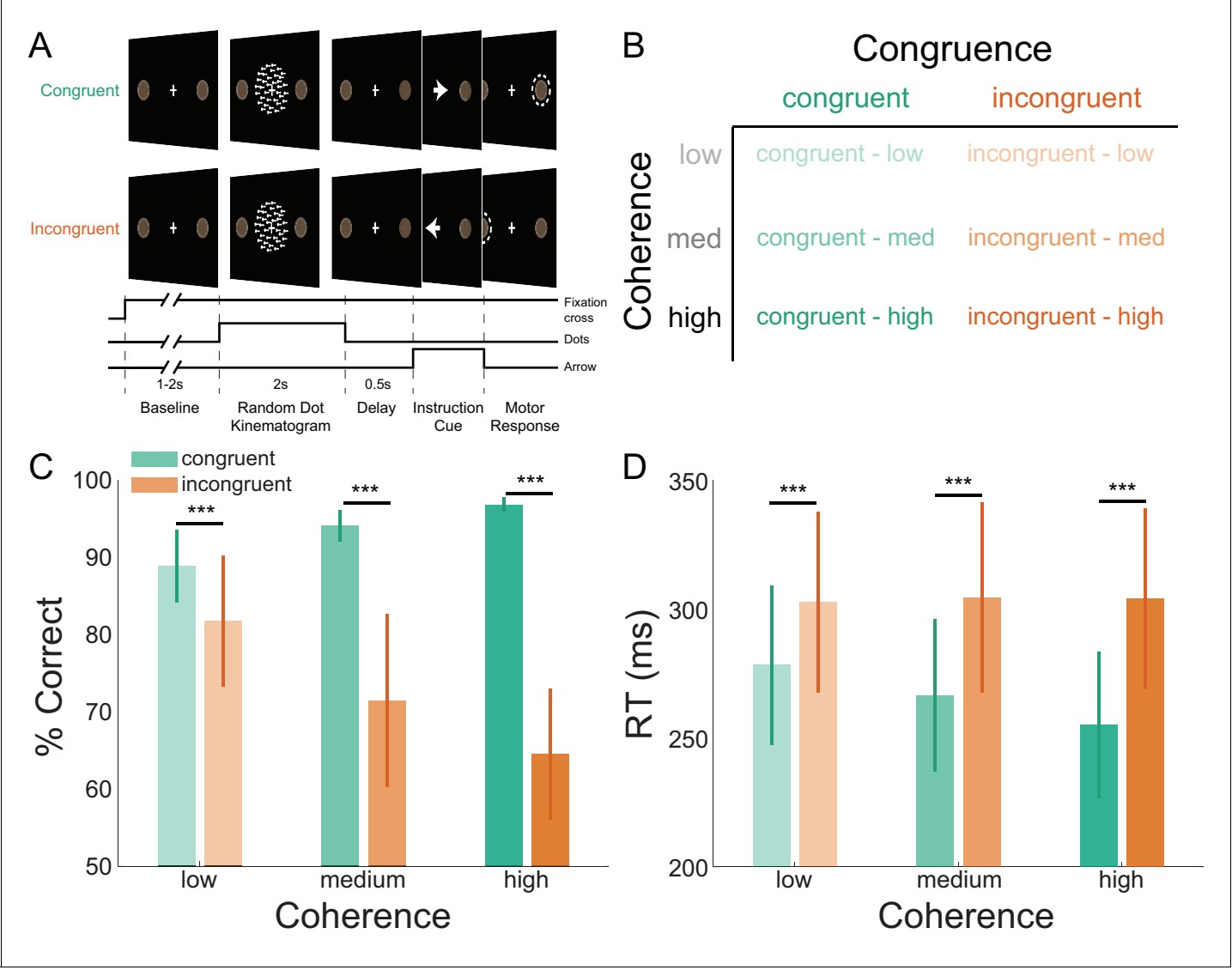

**Figure 1.** Task structure and participant behavior. (**A**) Each trial consisted of a fixation baseline (1 – 2 s), random dot kinematogram (RDK; 2 s), delay (0.5 s), and instruction cue interval, followed by a motor response (left/right button press) in response to the instruction cue (an arrow pointing in the direction of the required button press). During congruent trials the coherent motion of the RDK was in the same direction that the arrow pointed in the instruction cue, while in incongruent trials the instruction cue pointed in the opposite direction. (**B**) The task involved a factorial design, with three levels of motion coherence in the RDK and congruent or incongruent instruction cues. Most of the trials (70%) were congruent. (**C**) Mean accuracy over participants during each condition. Error bars denote the standard error. Accuracy increased with increasing coherence in congruent trials, and worsened with increasing coherence in incongruent trials. (**D**) The mean response time (RT) decreased with increasing coherence in congruent trials (***p<0.001). See *Figure 1* – source data for raw data.

DOI: https://doi.org/10.7554/eLife.33977.003

The following source data is available for figure 1:

**Source data 1.** Accuracy and response time data.

DOI: https://doi.org/10.7554/eLife.33977.004

ICC = 0.98; TF, button response, mean within-session ICC = 0.99, between-session ICC = 0.99). Across all subjects, the mean ICC for all task epochs and reproducibility measures was greater than 0.85 (topographic map, RDK, within-session ICC, M = 0.94, SD = 0.03, between-session ICC, M = 0.96, SD = 0.02; topographic map, instruction cue, within-session ICC, M = 0.97, SD = 0.03, between-session ICC, M = 0.98, SD = 0.02; topographic map, button response, within-session ICC, M = 0.96, SD = 0.03, between-session ICC, M = 0.95, SD = 0.06; ERF, RDK, within-session ICC,

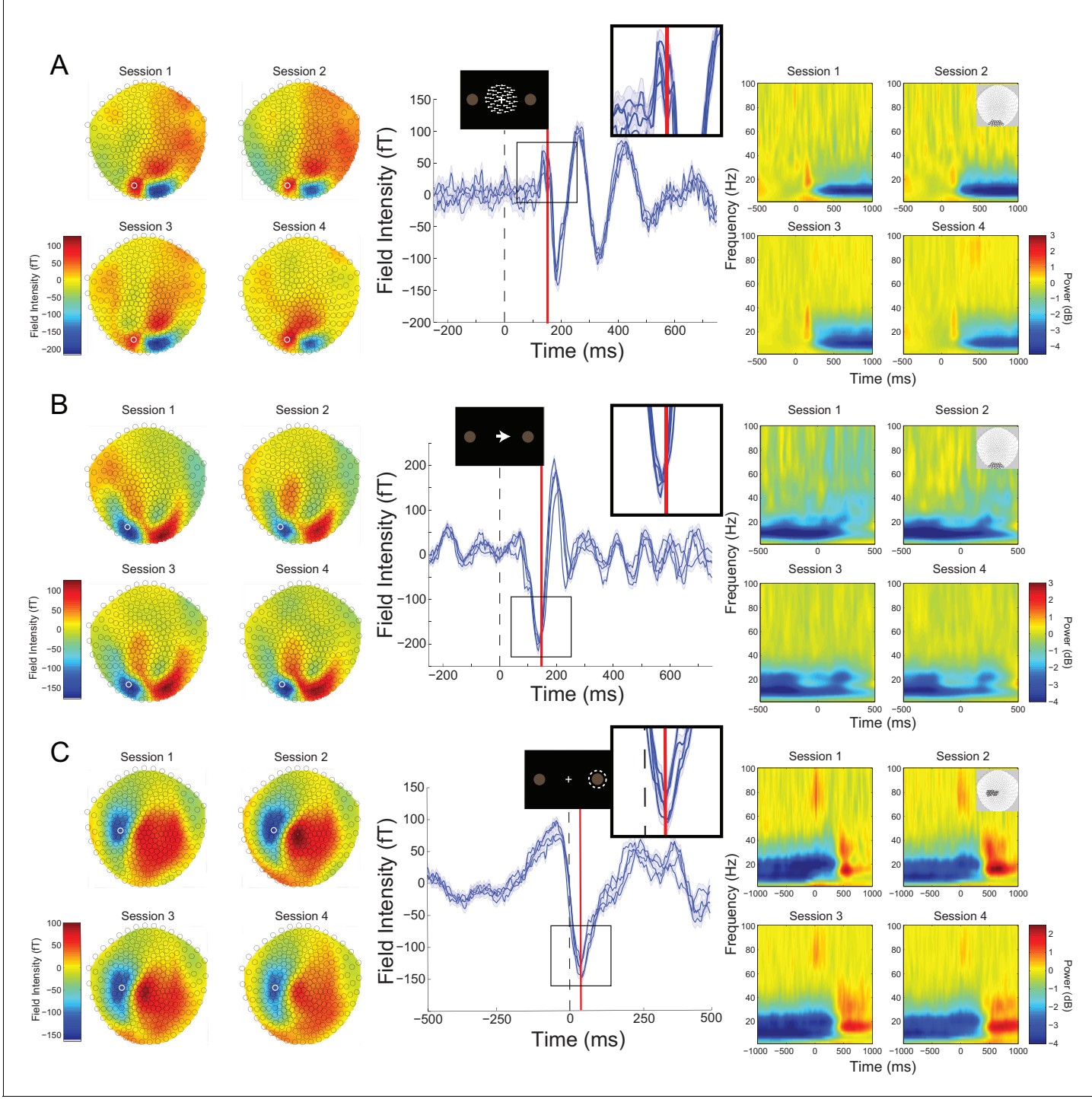

**Figure 2.** Cross-session reproducibility.Topographic maps (left column), event-related fields (ERFs, middle column), and time-frequency decompositions (right column). (**A**) aligned to the onset of the random dot kinematogram (RDK), (**B**) aligned to onset of the instruction cue, (**C**) aligned to the participant's response (button press). Data shown are for a single representative participant, with four sessions recorded on different days spaced at least a week apart (each including three, 15 min blocks with 180 trials per block). The white circles on the topographic maps denote the sensor from which the ERFs in the middle are recorded. Each blue line in the ERF plots represents a single session (average of 540 trials), with shading representing the standard error (within-session variability) and the red lines showing the time point that the topographic maps are plotted for (150 ms for the RDK and instruction cue, 35 ms for the response). The insets show a magnified view of the data plotted within the black square. The time-frequency decompositions are baseline corrected (RDK-aligned: [−500, 0 ms]; instruction cue-aligned: [−3 s, −2.5 s]; response-aligned: [−500 ms, 0 ms relative to the RDK]) and averaged over the sensors shown in the insets. See *Figure 2* – source data for raw data.

*Figure 2 continued on next page*

*Figure 2 continued*

DOI: https://doi.org/10.7554/eLife.33977.005

The following source data and figure supplement are available for figure 2:

**Source data 1.** Topographic, ERP, and time-frequency data for a representative participant across four recording sessions.
DOI: https://doi.org/10.7554/eLife.33977.007
**Figure supplement 1.** Within- and between-block fiducial coil variability.
DOI: https://doi.org/10.7554/eLife.33977.006

M = 0.88, SD = 0.08, between-session ICC, M = 0.94, SD = 0.05; ERF, instruction cue, within-session ICC, M = 0.93, SD = 0.03, between-session ICC, M = 0.94, SD = 0.03; ERF, button response, within-session ICC, M = 0.94, SD = 0.02, between-session ICC, M = 0.97, SD = 0.02; TF, RDK, within-session ICC, M = 0.95, SD = 0.03, between-session ICC, M = 0.97, SD = 0.01; TF, instruction cue, within-session ICC, M = 0.96, SD = 0.02, between-session ICC, M = 0.98, SD = 0.01; TF, button response, within-session ICC, M = 0.98, SD = 0.004, between-session ICC, M = 0.99, SD = 0.004).

## Task-related changes in low and high frequency activity

To address our main question about the laminar specificity of different frequency channels in human cortex, we first examined task-related low- and high-frequency activity from sensors overlying visual and sensorimotor cortices. Attention to visual stimuli is associated with decreases in alpha (*Hari, 1997*; *Sauseng et al., 2005*; *Yamagishi et al., 2005*; *Thut et al., 2006*; *Mazaheri et al., 2014*) and increases in gamma activity in visual cortex (*Müller et al., 1996*; *Fries et al., 2001*; *Busch et al., 2004*; *Hoogenboom et al., 2006*; *Muthukumaraswamy and Singh, 2013*). In line with previous research, sensors overlying the visual cortex revealed a significant decrease in alpha (7–13 Hz) and increase in gamma (60 – 90 Hz) power following the onset of the RDK and lasting for its duration (*Siegel et al., 2007*). In addition, we observed a burst of gamma activity following the onset of the instruction cue (*Figure 3A*; significant time-frequency windows marked, $p < 0.05$, Bonferroni corrected).

Motor responses are associated with a characteristic pattern of spectral activity in contralateral sensorimotor cortex, with a stereotypical decrease in average beta power prior to movement, followed by a rebound in average beta activity after the response. Moreover, a burst of gamma activity typically occurs around movement onset (*Pfurtscheller et al., 1996*; *Pfurtscheller and Neuper, 1997*; *Crone et al., 1998*; *Cheyne et al., 2008*; *Huo et al., 2010*; *Gaetz et al., 2011*). At the sensor-level, we indeed observed these classic average power changes, with a significant decrease in beta power (15 – 30 Hz) prior to and during the participant's response along with a subsequent rebound, and a burst of response-aligned gamma (60 – 90 Hz) activity (*Figure 3B*; significant time-frequency windows marked, $p < 0.05$, Bonferroni corrected). These signals are relevant for testing the proposed role of low and high frequency activity, respectively, for the following reasons. First, the average beta power decrease prior to movement has been linked to various processes related to the preparation and specification of movement (*Donner et al., 2009*; *Engel and Fries, 2010*; *Aron et al., 2016*; *Khanna and Carmena, 2017*; *Spitzer and Haegens, 2017*). Moreover, gamma bursts at movement onset are thought to originate from motor cortex, are effector-specific, and are thought to reflect processes related to the feedback control of movements (*Cheyne et al., 2008*; *Muthukumaraswamy, 2010*) and updating of motor plans (*Mehrkanoon et al., 2014*). However, we note that the proposed roles of pre- and post-movement beta and movement-onset gamma complicate the idea of these frequency channels conveying feedback and feedforward control, as seen in sensory cortices (*Bauer et al., 2014*; *Fontolan et al., 2014*; *van Kerkoerle et al., 2014*; *Bastos et al., 2015*; *Jensen et al., 2015*; *Michalareas et al., 2016*). This is because (a) the dynamics of beta activity occur both prior to and after the event (i.e., movement), whereas corresponding activity changes in sensory cortices are stimulus-driven; (b) the movement-onset gamma bursts have been linked to the initiation of movement and hence with descending corticospinal communication (*Cheyne et al., 2008*; *Cheyne and Ferrari, 2013*); and (c) motor cortex is agranular, which blurs the proposed laminar dissociation between feedback and feedforward information channels. This opens the possibility that movement-related beta and gamma activity may not be organized in the same lamina-specific manner as in sensory cortices. Alternatively, the same lamina-specific organization

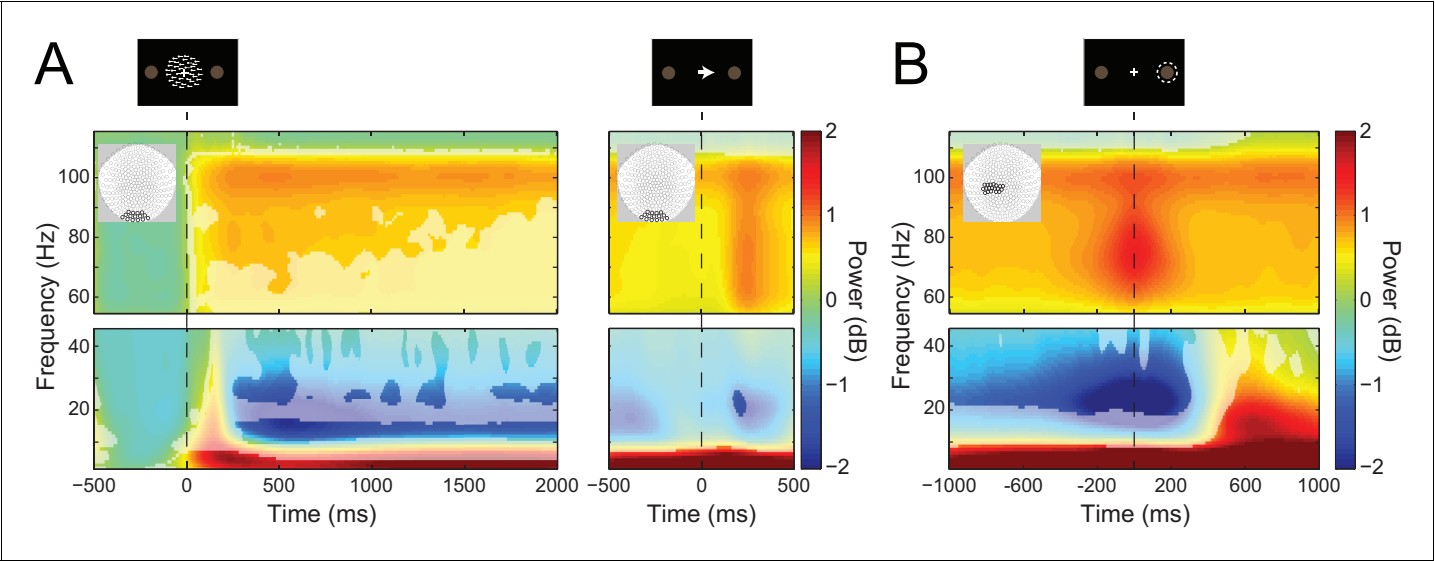

**Figure 3.** Visual and sensorimotor sensor-level activity. (**A**) Time-frequency representations of activity from sensors overlying visual cortex (shown in insets), aligned to the onset of the RDK (left) and the instruction cue (right). Data were baseline-corrected ([−500, 0 ms] relative to the onset of the RDK), and averaged over participants. Overlaid is a mask in which pixels where power is significantly changed from baseline are transparent, revealing the underlying time-frequency power. After the onset of the RDK, there is a sustained decrease in alpha, and increase in gamma activity, followed by a burst of gamma after the instruction cue. (**B**) Time-frequency representation of movement-related activity from sensors overlying contralateral sensorimotor cortex (shown in inset), aligned to the response, and baseline corrected ([−500, 0 ms] relative to the onset of the RDK), and averaged over participants. As in A, the mask overlaid reveals pixels with a significant change from baseline. There is a decrease in beta power prior to the motor response, followed by a beta rebound after the response, and a burst of gamma power aligned to the time of the response. See *Figure 3 – source data* for raw data.

DOI: https://doi.org/10.7554/eLife.33977.008

The following source data is available for figure 3:

**Source data 1.** Mean sensor-level time-frequency data for each participant.

DOI: https://doi.org/10.7554/eLife.33977.009

---

may have functional roles that are distinct from the proposed feedback and feedforward communication in sensory cortex.

## Low and high frequency activity localize to different cortical laminae

Having identified low- and high-frequency visual and sensorimotor signals at the sensor-level, we next asked whether these frequency channels indeed arise predominantly from deep or superficial cortical laminae. Localization of activity measured by MEG sensors requires accurate generative forward models which map from cortical source activity to measured sensor data (*Hillebrand and Barnes, 2002*; *Hillebrand and Barnes, 2003*; *Larson et al., 2014*; *Baillet, 2017*). We constructed a generative model for each participant based on a surface mesh that included both their white matter and pial surfaces, respectively (*Figure 4*, left column). This permits comparison of the estimated source activity for visual and sensorimotor activity on the white matter and pial surface. We infer a deep (white-matter boundary) laminar origin if the activity in a given frequency band is strongest on the white matter surface, and a superficial (pial surface) origin if this activity is strongest on the pial surface. For the purposes of comparison with invasive neural recordings, the deep laminae approximate infragranular cortical layers, and superficial laminae approximate supragranular layers.

The veracity of laminar inferences using this analysis is highly dependent on the accuracy of the white matter and pial surface segmentations. Imprecise surface reconstructions from standard 1 mm isotropic T1-weighted volumes result in coarse-grained meshes, which do not accurately capture the separation between the two surfaces, and thus are suboptimal for distinctions between deep and superficial laminae (*Figure 4—figure supplement 1*). We therefore extracted each surface from high-resolution (800 µm isotropic) MRI multi-parameter maps (*Carey et al., 2017*), allowing fine-grained segmentation of the white matter and pial surfaces (*Figure 4—figure supplement 1*).

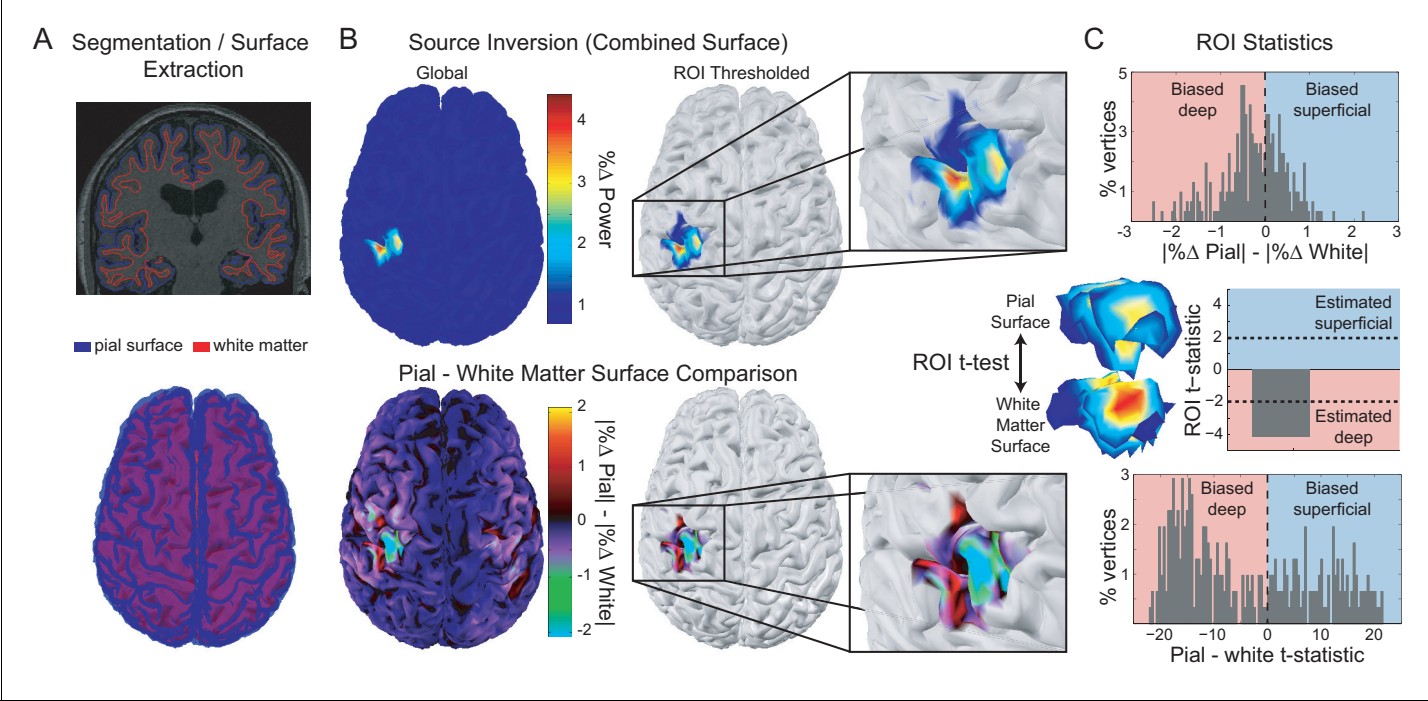

**Figure 4.** Laminar analysis approach. Pial and white matter surfaces are extracted from proton density and T1 weighted quantitative maps obtained from a multi-parameter mapping MRI protocol (A, top). A generative model combining both surfaces (A, bottom) is used to explain the measured sensor data, resulting in an estimate of the activity at every vertex on each surface (B, top left). The ROI analysis defined a region of interest by comparing the change in power in a particular frequency band during a time window of interest from a baseline time period (B, top right). The ROI includes all vertices in either surface in the 80th percentile (the top 20%) as well as corresponding vertices in the other surface. The unsigned fractional change in power from baseline on each surface was then compared within the ROI (B, bottom; C, top). Pairwise t-tests were performed between corresponding vertices on each surface within the ROI to examine the distribution of t-statistics (C, bottom), as well as on the mean unsigned fractional change in power within the ROI on each surface to obtain a single t-statistic which was negative if the greatest change in power occurred on the white matter surface, and positive if it occurred on the pial surface (C, middle).

DOI: https://doi.org/10.7554/eLife.33977.010

The following figure supplement is available for figure 4:

**Figure supplement 1.** FreeSurfer-extracted surfaces.
DOI: https://doi.org/10.7554/eLife.33977.011

For each low- and high-frequency visual and sensorimotor signal, the laminar analysis first calculated the unsigned fractional change in power from a baseline time window (i.e. power change from baseline divided by baseline power) on the vertices of each surface, and then compared this fractional power change between surfaces using paired t-tests over trials (*Figure 4C*, top). The resulting t-statistic was positive when the magnitude of the change in power was greater on the pial surface (superficial), and negative when the change was greater on the white matter surface (deep; *Figure 4C*, middle). To get a global measure of laminar specificity, we averaged this fractional change in power from baseline over the whole brain (all vertices) within each surface. For spatially localized laminar inference, we then identified regions of interest (ROIs) in each participant based on the mean frequency-specific change in power from a baseline time window on vertices from either surface (*Bonaiuto et al., 2018*). We compared two metrics for defining the ROIs: functionally defined (centered on the vertex with the peak mean difference in power), and anatomically-constrained (centered on the vertex with the peak mean power difference within the visual cortex bilaterally, or in the contralateral motor cortex). In addition to performing paired t-tests over trials using the unsigned fractional change in power from baseline averaged within ROIs, we also examined the distribution of t-statistics across vertices by performing a paired t-test across trials for each white matter/pial vertex pair (*Figure 4C*, bottom).

## Visual alpha and gamma have distinct laminar specific profiles

Based on *in vivo* laminar recordings in non-human primates (*Bollimunta et al., 2008*; *Bollimunta et al., 2011*; *Buffalo et al., 2011*; *Spaak et al., 2012*; *Xing et al., 2012*; *van Kerkoerle et al., 2014*; *Haegens et al., 2015*), we reasoned that changes in alpha activity following the RDK would predominantly arise from infragranular cortical layers. By contrast, changes in gamma activity following the RDK and instruction cue should be strongest in supragranular layers. Source reconstructions of the change in visual alpha activity following the onset of the RDK on the white matter and pial surfaces approximating the proposed laminar origins are shown in *Figure 5A* for an example participant over the whole brain and within the functionally defined ROI. Activity on both surfaces localized to posterior visual cortex bilaterally. When performing paired t-tests comparing corresponding vertices on the pial and white matter surfaces over all trials, the distribution of alpha activity was skewed toward the white matter surface, in line with the proposed infragranular origin. This bias was also observed within the functionally defined ROI. When averaging the change in power either over the whole brain, within a functionally-defined ROI, or an anatomically constrained ROI, the visual alpha activity of most participants was classified as originating from the white matter surface (global: $W(8)=0$, $p=0.008$, 8/8 participants, functional ROI: $W(8)=2$, $p=0.023$, 7/8 participants, anatomical ROI: $W(8)=16$, $p=0.844$, 5/8 participants; *Figure 5A*, right).

Conversely, the increase in visual gamma following the onset of the RDK and instruction cue was strongest on the pial surface (*Figure 5B,C*), as expected from invasive recordings (*Maier et al., 2010*; *Buffalo et al., 2011*; *Spaak et al., 2012*). Source reconstructions on the pial and the white matter surface for an example participant show the induced gamma activity over visual cortex (*Figure 5B,C*). For visual gamma, the distributions of t-statistics for pairwise vertex comparisons were skewed toward the pial surface, a finding that is compatible with a supragranular origin of high-frequency gamma activity. This was consistently observed for the global, functional, and anatomical ROI metrics (RDK gamma, global: $W(8)=35$, $p=0.016$, 7/8 participants; RDK gamma, functional ROI: $W(8)=35$, $p=0.016$, 7/8 participants; RDK gamma, anatomical ROI: $W(8)=35$, $p=0.016$, 7/8 participants; instruction cue gamma, global: $W(8)=35$, $p=0.016$, 7/8 participants; instruction cue gamma, functional ROI: $W(8)=35$, $p=0.016$, 7/8 participants; instruction cue gamma, anatomical ROI: $W(8)=28$, $p=0.195$, 5/8 participants).

## Sensorimotor beta and gamma originate from distinct cortical laminae

The above results provide novel support for distinct anatomical pathways through which different frequency channels contribute to inter-areal communication within visual cortices. We next addressed whether this laminar specificity of different frequency channels was common to other portions of cortex, specifically the movement-related changes originating from sensorimotor cortex.

Cortical regions vary in terms of thickness (*Fischl and Dale, 2000*; *Jones et al., 2000*; *MacDonald et al., 2000*; *Kabani et al., 2001*; *Lerch and Evans, 2005*), as a result of inter-regional variation in cortical folding and laminar morphology (*Barbas and Pandya, 1989*; *Matelli et al., 1991*; *Rajkowska and Goldman-Rakic, 1995*; *Hilgetag and Barbas, 2006*). Moreover, the distinction between feedback and feedforward cortical processing channels may be less clear for motor cortex, which is agranular (missing layer IV) and projects directly to the spinal cord. Supporting this argument, motor gamma bursts are closely tied to movement onset, and have been linked to movement execution and feedback control (*Cheyne et al., 2008*; *Cheyne and Ferrari, 2013*).

While frequency-specific activity thus occurs throughout cortex, the laminar distribution of different frequency channels may differ across different levels in the cortical hierarchy. Because MEG is predominantly sensitive to the synchronous activity of large populations of pyramidal cells, it is likely that different laminar microcircuits could give rise to the same measurable MEG signals (*Cohen, 2017*). Alternatively, if the layer specificity of low and high frequency activity is indeed a general organizing principle throughout cortex, the pre-movement beta decrease and post-movement rebound ought to originate from infragranular cortical layers, whereas the movement-related gamma increase ought to be strongest in supragranular layers. Moreover, the ability of MEG to accurately segregate deep from superficial laminar source activity may vary throughout cortex, a possibility we have previously explored in simulation (*Bonaiuto et al., 2018*).

To explore this possibility empirically, we analyzed two task-related modulations of sensorimotor beta activity: the decrease in beta power following the onset of the RDK and prior to the motor

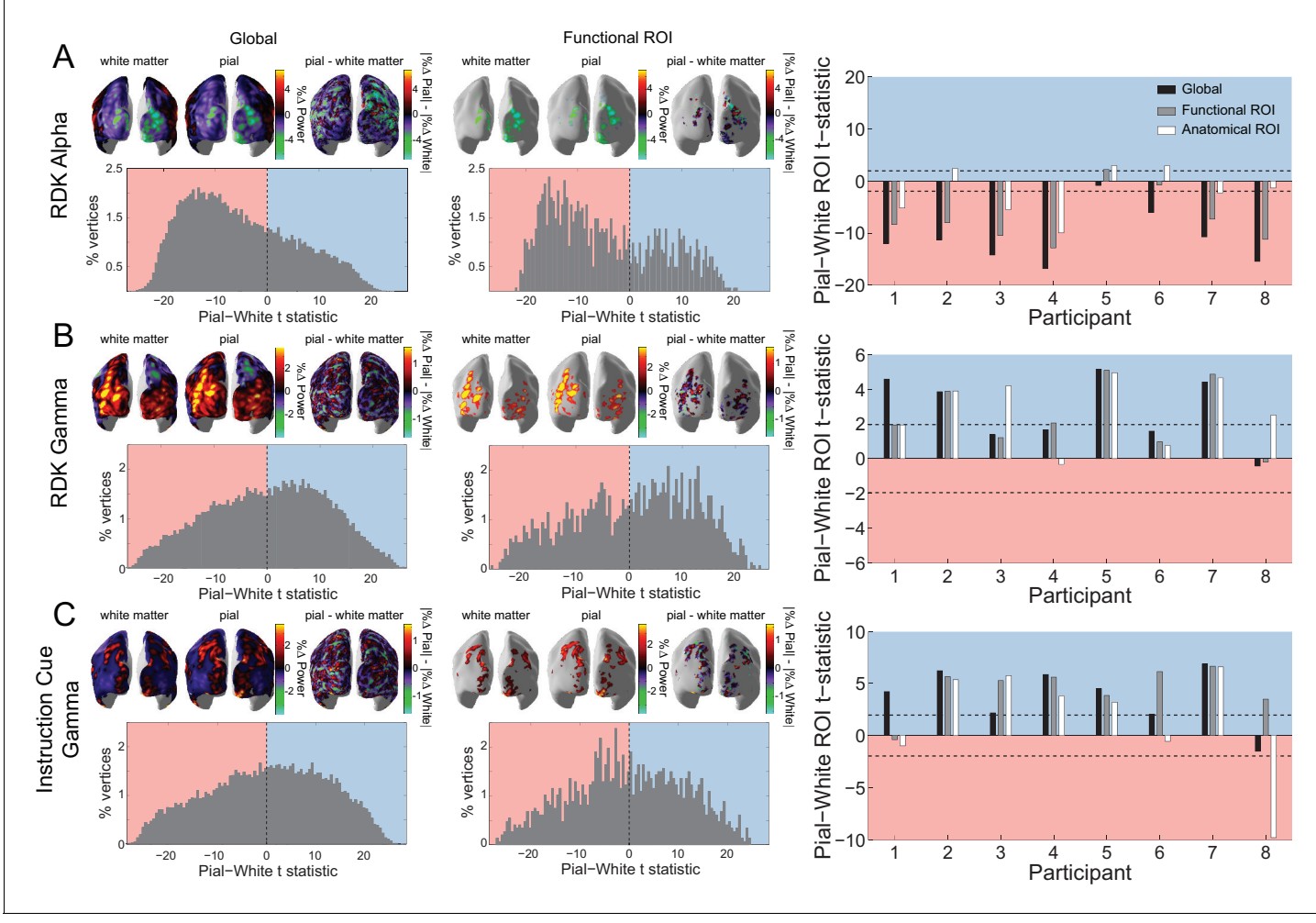

**Figure 5.** Laminar specificity of visual alpha and gamma. (A) Estimated changes in alpha power (7 – 13 Hz) from baseline on the white matter and pial surface, and the difference in the unsigned fractional change in power (pial – white matter) following the onset of the random dot kinematogram (RDK), over the whole brain (global) and within a functionally defined region of interest (ROI). Histograms show the distribution of t-statistics comparing the fractional change in power from baseline between corresponding pial and white matter surface vertices over the whole brain, or within the ROI. Negative t-statistics indicate a bias toward the white matter surface, and positive t-statistics indicate a pial bias. The bar plots show the t-statistics comparing the fractional change in power from baseline between the pial and white matter surfaces averaged within the ROIs, over all participants. T-statistics for the whole brain (black bars), functionally defined (grey bars), and anatomically constrained (white bars) ROIs are shown (red = biased toward the white matter surface, blue = biased pial). Dashed lines indicate the threshold for single participant statistical significance. (B) As in A, for gamma (60 – 90 Hz) power following the RDK. C) As in A and B, for gamma (60 – 90 Hz) power following the instruction cue. See *Figure 5* – source data for raw data.

DOI: https://doi.org/10.7554/eLife.33977.012

The following source data and figure supplements are available for figure 5:

**Source data 1.** Laminar comparison data for visual alpha (RDK) and gamma (RDK and instruction cue).
DOI: https://doi.org/10.7554/eLife.33977.026
**Figure supplement 1.** Sensor shuffling biases visual and sensorimotor laminar specificity to the pial surface.
DOI: https://doi.org/10.7554/eLife.33977.013
**Figure supplement 2.** Adding coregistration error biases visual and sensorimotor laminar specificity to the pial surface.
DOI: https://doi.org/10.7554/eLife.33977.014
**Figure supplement 3.** Visual and sensorimotor laminar specificity compared to sensor shuffled data.
DOI: https://doi.org/10.7554/eLife.33977.015
**Figure supplement 4.** Laminar inference is a function of the number of trials.
DOI: https://doi.org/10.7554/eLife.33977.016
**Figure supplement 5.** Laminar localization as a function of SNR.
DOI: https://doi.org/10.7554/eLife.33977.017

*Figure 5 continued*

**Figure supplement 6.** Laminar preference scales with the difference in pial and white matter lead field strength.
DOI: https://doi.org/10.7554/eLife.33977.018
**Figure supplement 7.** The relationship between pial and white matter lead field strength and laminar preference is the same across frequency bands.
DOI: https://doi.org/10.7554/eLife.33977.019
**Figure supplement 8.** Laminar preference does not relate to the difference in vertex depth.
DOI: https://doi.org/10.7554/eLife.33977.020
**Figure supplement 9.** Laminar localization does not reverse for vertex pairs in which the white matter surface is closer to the scalp than the pial surface.
DOI: https://doi.org/10.7554/eLife.33977.021
**Figure supplement 10.** Visual and sensorimotor laminar specificity does not change after controlling for distance to the scalp.
DOI: https://doi.org/10.7554/eLife.33977.022
**Figure supplement 11.** The relationship between patch size estimates and laminar bias.
DOI: https://doi.org/10.7554/eLife.33977.023
**Figure supplement 12.** Sensor covariance is similar across frequency bands.
DOI: https://doi.org/10.7554/eLife.33977.024
**Figure supplement 13.** Laminar localization is not affected by regularization or sensor covariance.
DOI: https://doi.org/10.7554/eLife.33977.025

response, and the post-movement beta rebound (*Salmelin et al., 1995*; *Pfurtscheller et al., 1996*; *Cassim et al., 2001*; *Jurkiewicz et al., 2006*; *Parkes et al., 2006*). Both signals localized to the left sensorimotor cortex (contralateral to the hand used to indicate the response; *Figure 6A,B*). For both epochs, the signal was strongest on the white matter surface, as evidenced by the white matter skews in the global and functional ROI t-statistics (*Figure 6*). This result held for all but one participant at the single participant level, and overall at the group level (beta decrease, global: $W(8)=0$, $p=0.008$; beta decrease, functional ROI: $W(8)=6$, $p=0.109$; beta decrease, anatomical ROI: $W(8)=0$, $p=0.008$; beta rebound, global: $W(8)=1$, $p=0.016$; beta rebound, functional ROI: $W(8)=2$, $p=0.023$; beta rebound, anatomical ROI: $W(8)=0$, $p=0.008$).

Turning to the burst of gamma aligned with the onset of the movement and localized to the same patch of left sensorimotor cortex (*Figure 6C*), we found that this signal was strongest on the pial surface (global: $W(8)=35$, $p=0.016$, 7/8 participants; functional ROI: $W(8)=33$, $p=0.039$, 6/8 participants; anatomical ROI: $W(8)=31$, $p=0.078$, 6/8 participants).

## Laminar discrimination is disrupted by adding spatial and temporal noise

We then conducted several control analyses to ascertain the robustness of our findings: i) shuffling the location of the sensors (effectively assigning the data from one sensor to another), ii) simulating increased co-registration error, and iii) decreasing effective SNR by using only a random subset of the trials for each participant or adding white noise at the sensor level.

Shuffling the position of the sensors destroys any correspondence between the anatomy and the sensor data. Added co-registration error simulates the effect of between-session spatial uncertainty arising from head movement and inaccuracies of the forward model typically experienced without head-casts (*Uutela et al., 2001*; *Hillebrand and Barnes, 2003*; *Hillebrand and Barnes, 2011*; *Medvedovsky et al., 2007*; *Troebinger et al., 2014a*; *Meyer et al., 2017b*). For both control analyses, all signals now localized to the pial surface (*Figure 5—figure supplements 1* and *2*), suggesting that the laminar discrimination between low- and high-frequency signals in our main analyses relies on precise anatomical models. We additionally re-ran our main laminar comparisons, now testing against the null hypothesis that the difference (pial-white) of the unsigned fractional change in power from baseline within an ROI is equal to the value obtained from sensor shuffling (rather than the default null hypothesis that the difference is zero). This revealed the same pattern of laminar bias, with visual alpha and sensorimotor beta activity localizing to deep laminae, and visual and sensorimotor gamma localizing superficially (*Figure 5—figure supplement 3*).

On average, the magnitude of the t-statistics in our global and ROI analyses increased with the number of trials used in the analysis, with more trials required for gamma signals to reach significance (*Figure 5—figure supplement 4*). One concern was that the effects could be driven by the absolute power of our signal, in that higher power signals always localize to deeper structures.

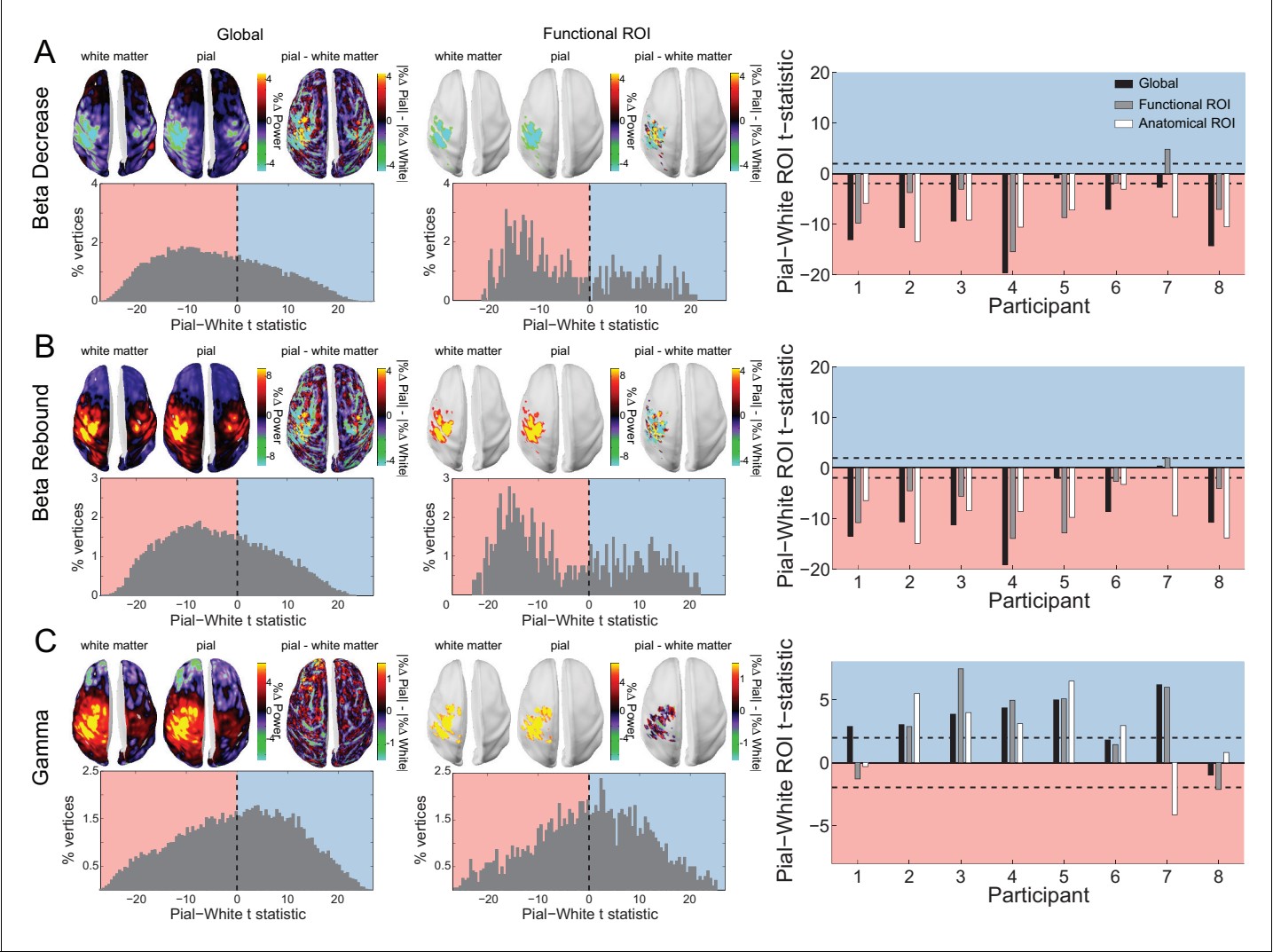

**Figure 6.** Laminar specificity of sensorimotor beta and gamma. As in **Figure 5**, for (**A**) the beta (15 – 30 Hz) decrease prior to the response, (**B**) beta (15 – 30 Hz) rebound following the response, and (**C**) gamma (60 – 90 Hz) power change from baseline during the response. In the histograms and bar plots, positive and negative values indicate a bias towards the superficial and deeper cortical laminae, respectively. The dashed lines indicate single participant-level significance thresholds. The black, grey, and white bars indicate statistics based on regions of interest comprising the whole brain, functional and anatomically-constrained ROIs, respectively. See **Figure 6** – source data for raw data.

DOI: https://doi.org/10.7554/eLife.33977.027

The following source data is available for figure 6:

**Source data 1.** Laminar comparison data for sensorimotor beta decrease, beta rebound, and gamma.

DOI: https://doi.org/10.7554/eLife.33977.028

Importantly, however, regardless of the SNR, the trivial superficial bias of the shuffled sensor models was weaker than that of the unshuffled sensor models, both within the functionally defined, and the anatomically constrained ROIs (**Figure 5—figure supplement 4**). Moreover, whereas adding progressively more white noise to the sensor level data steadily increased the superficial bias of visual alpha and sensorimotor beta until a point of saturation was reached, the change in the laminar bias of visual and sensorimotor gamma saturated at a much lower noise level and became unstable for some subjects and contrasts, flipping from a superficial to a deep bias (**Figure 5—figure supplement 5**). If the superficial localization of gamma were a trivial consequence of low SNR we would have expected the addition of noise to have little effect (i.e. the curves would be already at

saturation point). We would also not have expected the inference to flip in the opposite direction as noise was added (implying that adding this noise obscured some meaningful gamma signal).

## Influence of cortical anatomy on laminar discrimination

One concern is that our results could have been driven by the relative distance of a given vertex pair from the scalp surface (and hence the MEG sensors). The difference in lead field strength is a parsimonious quantity to address this concern, because it depends on both distance to the sensors as well as the orientation of the cortical surface. This analysis revealed a correlation between relative pial/white matter lead field strength and laminar preference (*Figure 5—figure supplement 6A*), with a tendency to localize activity to the vertex with the stronger lead field. This, in turn, raises the issue of whether the vertices contributing to the laminar bias we observed were simply those with the strongest lead field.

However, and importantly for the main findings of this paper, this relationship was not frequency-specific (*Figure 5—figure supplement 7*), and even when pial and white matter vertices were matched for lead field strength (within 1% of the overall range), a clear dissociation between low and high frequency signals was still evident at the single participant level (*Figure 5—figure supplement 6B*). Low frequency activity was consistently localized toward deep layers, whereas for this sub-sample of vertices the high frequency activity showed no layer bias. We observed similar effects across two separate brain regions and three task epochs. These analyses were corroborated by analyses showing that the relative distance to the scalp surface did not trivially determine laminar preference (*Figure 5—figure supplement 8*), and that comparing ROIs containing only vertex pairs in which the white matter vertex is closer to the scalp than the pial vertex resulted in a similar pattern of laminar localization (*Figure 5—figure supplement 9*).

There appears to be a relationship between cortical folding and laminar bias, as evident in the cortical distribution of the difference in the unsigned fractional change in power (pial – white matter) over the whole brain (*Figures 4* and *5*). This manifests as a deep layer bias on the gyral crowns, and a superficial bias in the sulcal fundi. We controlled for this bias by analyzing the residuals of a regression predicting the difference in the unsigned fractional change in power (pia – white) from the square root of the distance to the scalp (averaged over pial and white matter vertex pairs). Crucially, this analysis did not change the laminar localization of low and high frequency signals (*Figure 5—figure supplement 10*).

Finally, as discussed previously (*Troebinger et al., 2014a*; *Bonaiuto et al., 2018*), over- or underestimation of source patch sizes can bias laminar results. We tested a representative participant using a range of patch sizes (from 2.5 to 20 mm FWHM). Regardless of patch size, the low frequency signals were consistently estimated to originate from deeper laminae. Generally, we found that smaller patch sizes tend to push our estimates in a superficial direction whereas large patch sizes tended to introduce a deep laminar bias; this had the greatest effect on the high-frequency estimates. However, at the optimal patch size (as determined by free energy comparison of combined pial/white matter source inversions), low frequency activity localized to deep laminae and high frequency activity to superficial laminae (*Figure 5—figure supplement 11*). Based on invasive recordings (*Leopold and Logothetis, 2003*), we had expected patch size to decrease monotonically with frequency, but did not observe such a relationship (*Figure 5—figure supplement 11*). We acknowledge, however, that our models are based on homogeneous Gaussian patches of activity, which therefore may not be realistic.

## Superficial visual gamma scales with cue congruence

Next, we asked whether the observed low and high-frequency lamina-specific activity in visual and sensorimotor cortex dynamically varied with task demands in line with proposals about their role in feedback and feedforward message passing (*von Stein et al., 2000*; *Fries, 2005*; *Fries, 2015*; *Friston and Kiebel, 2009*; *Wang, 2010*; *Jensen and Mazaheri, 2010*; *Donner and Siegel, 2011*; *Arnal and Giraud, 2012*; *Bastos et al., 2012*; *Adams et al., 2013*; *Jensen et al., 2015*; *Stephan et al., 2017*). This would provide additional indirect support for the idea that communication in hierarchical cortical networks is organized through distinct frequency channels along distinct anatomical pathways, to orchestrate top-down and bottom-up control.

In our task, the direction of the instruction cue was congruent with the motion coherence direction in the RDK during most trials (70%). As such, if the direction of motion coherence is to the left, the instruction cue will most likely be a leftward arrow. Gamma activity increases in sensory areas during presentation of unexpected stimuli (*Gurtubay et al., 2001*; *Arnal et al., 2011*; *Todorovic et al., 2011*), and therefore we expected visual gamma activity in supragranular layers to be greater following incongruent instruction cues than after congruent cues. Indeed, the increase in visual gamma on the pial surface following the onset of the instruction cue was greater in incongruent compared to congruent trials (*W*(8)=0, *p*=0.008; 8/8 participants; incongruent % change from baseline - congruent % change from baseline M = 1.64%, SD = 2.34%; *Figure 7*).

## Deep sensorimotor beta scales with RDK motion coherence and cue congruence

Changes in sensorimotor beta power during response preparation predict forthcoming motor responses (*Donner et al., 2009*; *Haegens et al., 2011*; *de Lange et al., 2013*), whereas the magnitude of sensorimotor beta rebound is attenuated by movement errors (*Tan et al., 2014*; *Tan et al., 2016*; *Torrecillos et al., 2015*). We therefore predicted that, in infragranular layers, the decrease in sensorimotor beta would scale with the motion coherence of the RDK, and the magnitude of the beta rebound would be decreased during incongruent trials when the prepared movement has to be changed in order to make a correct response.

The behavioral results presented thus far suggest that participants accumulated perceptual evidence from the RDK in order to prepare their response prior to the onset of the instruction cue. This preparation was accompanied by a reduction in beta power in the sensorimotor cortex contralateral

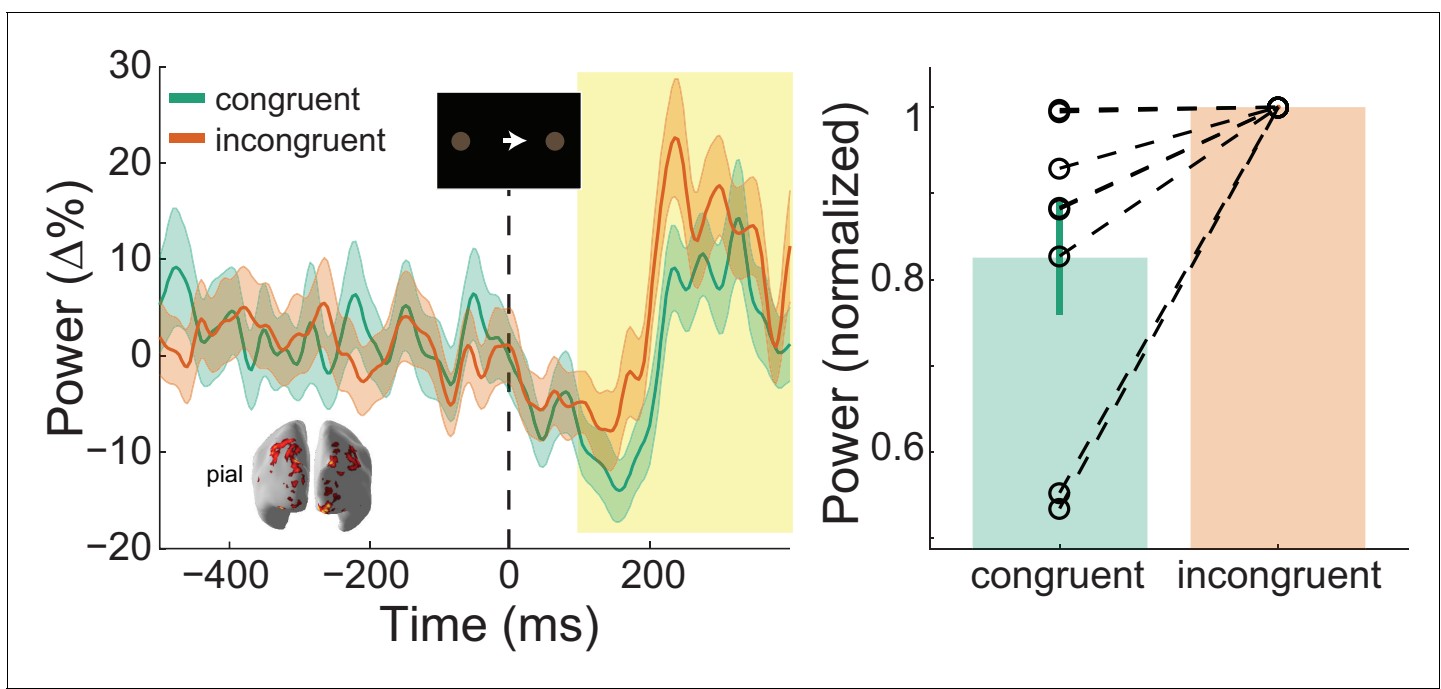

**Figure 7.** : Visual gamma activity modulation by task condition. Visual gamma activity following the onset of the instruction stimulus within the functionally defined ROI of an example participant (left), and averaged within the time window represented by the shaded yellow rectangle for all participants (right). Each dashed line on the right shows the change in normalized values for the different conditions for each participant. The bar height represents the mean normalized change in gamma power, and the error bars denote the standard error. Visual gamma activity is stronger following the onset of the instruction cue when it is incongruent to the direction of the coherent motion in the random dot kinematogram (RDK). See *Figure 7* – source data for raw data.

DOI: https://doi.org/10.7554/eLife.33977.029

The following source data is available for figure 7:

**Source data 1.** Condition comparison data for visual gamma (instruction cue).
DOI: https://doi.org/10.7554/eLife.33977.030

to the hand used to indicate the response (*Figure 6A*). This beta decrease began from the onset of the RDK and was more pronounced with increasing coherence, demonstrating a significant effect of

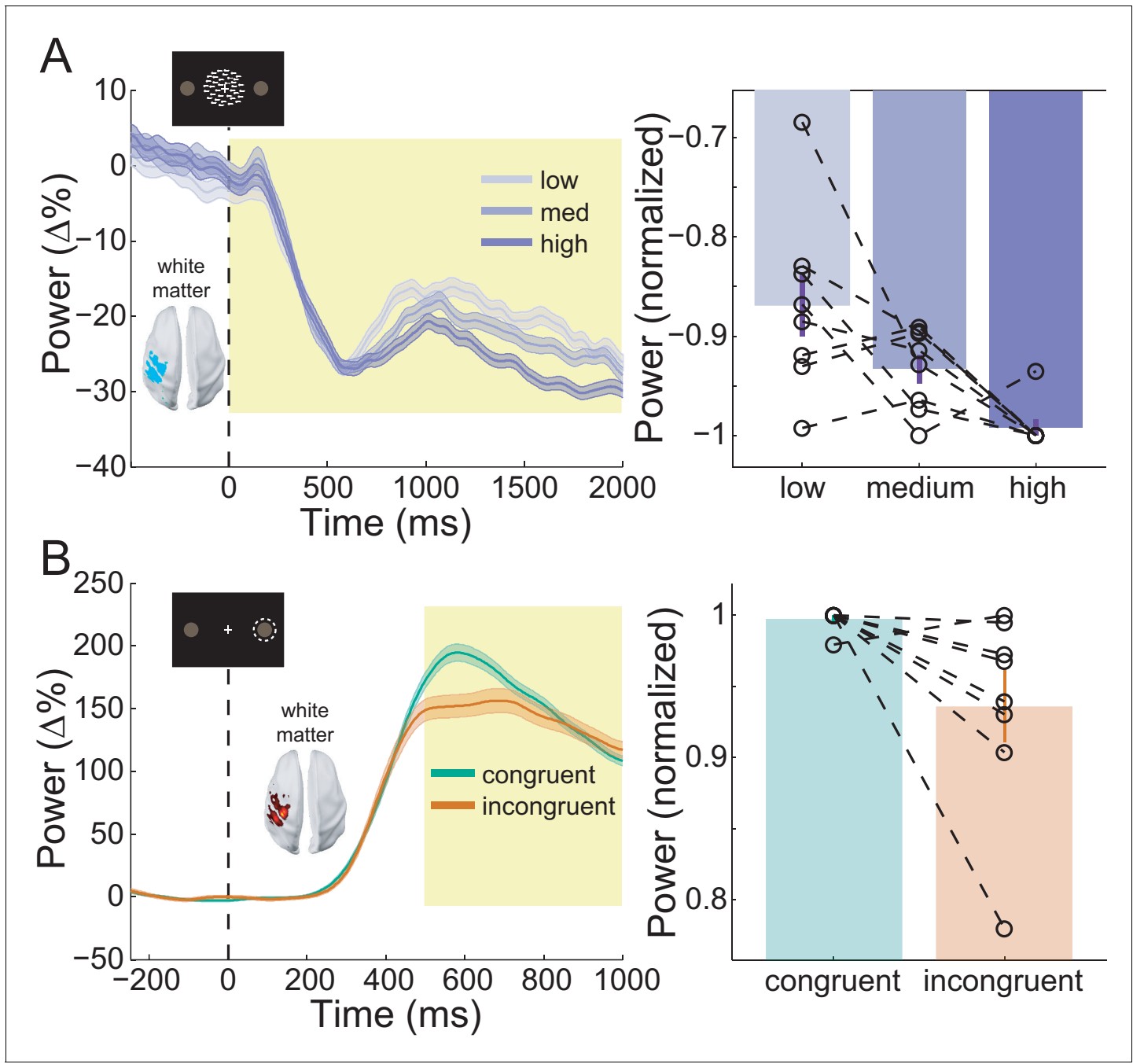

**Figure 8.** Sensorimotor beta activity modulated by task condition. (**A**) Beta decrease following the onset of the random dot kinematogram (RDK) within the functionally defined ROI of an example participant over the duration of the RDK (left), and averaged over this duration for all participants (right). The bar height represents the mean normalized change in beta power, and the error bars denote the standard error. The beta decrease becomes more pronounced with increasing coherence. (**B**) As in A, for beta rebound following the response and averaged within the time window shown by the shaded yellow rectangle. Beta rebound is stronger following responses in congruent trials. See *Figure 8* – source data for raw data.
DOI: https://doi.org/10.7554/eLife.33977.031

The following source data is available for figure 8:

**Source data 1.** Condition comparison data for sensorimotor beta decrease and beta rebound.
DOI: https://doi.org/10.7554/eLife.33977.032

coherence on the white matter surface (*Figure 8A*; $X^2(2)$=9.75, $p$=0.008), with beta during high coherence trials significantly lower than during low coherence trials (8/8 participants; $t(7)$=-3.496, $p$=0.033; low % change from baseline – high % change from baseline M = 2.42%, SD = 1.96%). Following the response, there was an increase in beta in contralateral sensorimotor cortex (beta rebound) which was greater in congruent, compared to incongruent trials on the white matter surface (*Figure 8B*; $W(8)$=34, $p$=0.023; 7/8 participants, congruent % change from baseline - incongruent % change from baseline M = 5.13%, SD = 5.19%). In other words, the beta rebound was greatest when the cued response matched the prepared response.

## Discussion

We here provide non-invasive evidence from human MEG recordings that low frequency channels of activity localize predominantly to deep laminae, and high frequency activity channels localize more superficially, in both visual and sensorimotor cortex. Through the use of novel MEG head-cast technology (*Troebinger et al., 2014b*; *Meyer et al., 2017a*) and spatially and temporally resolved laminar analyses (*Troebinger et al., 2014a*; *Bonaiuto et al., 2018*), our results provide non-invasive support for layer- and frequency-specific accounts of hierarchical cortical organization in humans.

### Lamina-resolved MEG of distinct frequency channels in human visual and sensorimotor cortex

In this study, we sought to address recent proposals about the role of distinct frequency channels of activity in hierarchical processing (*Fries, 2005*; *Fries, 2015*; *Friston and Kiebel, 2009*; *Wang, 2010*; *Jensen and Mazaheri, 2010*; *Donner and Siegel, 2011*; *Arnal and Giraud, 2012*; *Bastos et al., 2012*; *Adams et al., 2013*; *Jensen et al., 2015*; *Stephan et al., 2017*; though see *Haegens et al., 2015*; *Halgren et al., 2017*). According to these proposals, ascending (bottom-up) and descending (top-down) information processing occurs through distinct anatomical and frequency-specific channels. Whereas bottom-up information is conveyed via high frequency activity in supragranular layers, top-down information is associated with low frequency activity in infragranular layers. Currently, few studies in humans have tested these proposals, often on indirect grounds (*Koopmans et al., 2010*; *Olman et al., 2012*; *Fontolan et al., 2014*; *Kok et al., 2016*; *Michalareas et al., 2016*; *Scheeringa and Fries, 2017*). Moreover, these studies have generally focused on sensory systems, whereas here we sought to establish the generalizability of these proposals across cortex, and therefore additionally focused on agranular sensorimotor cortex.

When interpreting our results, it is therefore important to consider whether or not it is principally possible to achieve the spatial precision needed to distinguish deep versus superficial laminae activity with MEG. As MEG is a direct measure of neural activity, its spatial precision is, in principle, only limited by the signal-to-noise ratio with which data can be recorded, and the analysis techniques used to perform source localization (*Hillebrand and Barnes, 2002*; *Hillebrand and Barnes, 2003*; *Hillebrand and Barnes, 2011*; *Brookes et al., 2010*; *López et al., 2012*; *Troebinger et al., 2014b*; *Meyer et al., 2017a*; *Bonaiuto et al., 2018*).

Notably, in addition to theoretical considerations that a distinction of sources as close as 2 – 3 mm with MEG is feasible, recent MEG work on the retinotopic organization of visually induced activity provides empirical support for this precision (*Nasiotis et al., 2017*). These authors quantified the smallest detectable change in source location elicited by a shift in the position of a visual stimulus, which was as low as 1 mm.

### Low and high frequency channels localize to deep and more superficial cortical laminae across visual and sensorimotor cortex

We found that low frequency activity (alpha, 7 – 13 Hz; and beta, 15 – 30 Hz) predominately originated from deep cortical laminae, and high frequency activity (gamma, 60 – 90 Hz) from more superficial laminae in both visual and sensorimotor cortex. Our analysis included a built-in control: visually induced gamma after both the RDK and the instruction cue localized superficially, reinforcing the proposal that visual gamma generally predominates from superficial laminae. Moreover, laminar specificity was abolished by shuffling the sensors (*Figure 5—figure supplement 1*) or introducing co-registration error (*Figure 5—figure supplement 2*), underlining the need for spatially precise anatomical data and MEG recordings. Importantly, the laminar bias of both low and high frequency

signals increased monotonically as the number of trials included in the analysis increased, but this effect was weaker when the sensors were shuffled (*Figure 5—figure supplement 4*), and the superficial bias of all signals increased until saturation with the addition of increasing levels of white noise, but high frequency signals saturated at much lower noise levels and the superficial bias became unstable with increasing noise levels (*Figure 5—figure supplement 5*). These results suggest that the more superficial localization of gamma signals was not simply due to a trivial relationship between laminar bias and SNR.

Additionally, we established that our results were not simply driven by the relative strength of the pial and white matter surface lead fields. While we found a correlation between relative lead field strength and laminar preference (*Figure 5—figure supplement 6A*), this relationship was constant across frequency bands (*Figure 5—figure supplement 7*), and the laminar dissociation held at the single participant level when considering only vertex pairs matched for lead field strength (*Figure 5—figure supplement 6B*). Moreover, the deep laminar preference of low frequency signals was preserved even when considering only vertex pairs where the white matter vertex was closer to the scalp than the pial vertex (*Figure 5—figure supplement 9*). These results suggest that our main analyses were sensitive to the likely source of low- and high-frequency signals (rather being simply dependent on the relative magnitude of the influence of source activity from the pial versus white matter surface on the MEG sensors). However, while the slope of the relationship between relative lead field strength and laminar preference was constant across frequency bands, for gamma signals, this regression fit had an offset of approximately zero (*Figure 5—figure supplement 7*). Moreover, the laminar preference of sensorimotor gamma within the anatomically constrained ROIs reversed when considering only vertex pairs in which the white matter vertex was closest to the scalp. Given these issues, the conservative conclusion would be that visual and sensorimotor gamma localize *more* superficially than visual alpha and sensorimotor beta.

One possible confound in our analysis is the estimate of sensor noise. We assumed this to be diagonal. However subsequent tests, based on independent data recorded during a similar time-period, showed off-diagonal structure (*Figure 5—figure supplement 12*). Although this structure was the same across frequency bands it will have affected the free energy optimization stage. However, when using a sensor covariance matrix based on empty room noise measurements, the same pattern of laminar preference was observed (*Figure 5—figure supplement 13C*).

The localization of alpha activity to predominately deep laminae of visual cortex is in line with evidence from depth electrode recordings in visual areas of the non-human primate brain (*Maier et al., 2010*; *Buffalo et al., 2011*; *Spaak et al., 2012*; *Xing et al., 2012*; *Smith et al., 2013*; *van Kerkoerle et al., 2014*). Several studies have found alpha generators in both infra- and supragranular layers in primary sensory areas (*Bollimunta et al., 2008*; *Bollimunta et al., 2011*; *Haegens et al., 2015*), and it has been suggested that this discrepancy is due to a contamination of infragranular layer LFP signals by volume conduction from strong alpha generators in supragranular layers (*Haegens et al., 2015*; *Halgren et al., 2017*). This is unlikely to apply to the results presented here as this type of laminar MEG analysis is biased toward superficial laminae when SNR is low (*Bonaiuto et al., 2018*). However, this analysis is binary (deep or superficial) and will be biased toward the region of highest power change, even if the true source distribution populates multiple depths (*Bonaiuto et al., 2018*).

We found that gamma activity was strongest in more superficial sources, confirming invasive recordings showing gamma activity arising predominantly from supragranular layers in visual cortex (*Buffalo et al., 2011*; *Spaak et al., 2012*; *Xing et al., 2012*; *Smith et al., 2013*; *van Kerkoerle et al., 2014*; but see *Nandy et al., 2017*). The mechanisms underlying the generation of gamma activity are diverse across the cortex (*Buzsáki and Wang, 2012*), but commonly involve reciprocal connections between pyramidal cells and interneurons, or between interneurons (*Tiesinga and Sejnowski, 2009*; *Whittington et al., 2011*). The local recurrent connections necessary for such reciprocal interactions are most numerous in supragranular layers (*Buzsáki and Wang, 2012*), as are fast-spiking interneurons which play a critical role in generating gamma activity (*Cardin et al., 2009*; *Sohal et al., 2009*; *Carlén et al., 2012*).

It is hypothesized that the laminar segregation of frequency channels is a common organizing principle across the cortical hierarchy (*Wang, 2010*; *Arnal and Giraud, 2012*; *Bastos et al., 2012*; *Fries, 2015*). However, most evidence for this claim comes from depth electrode recordings in primary sensory areas, with the vast majority in visual cortical regions (*Buffalo et al., 2011*;

*Spaak et al., 2012*; *Xing et al., 2012*; *Smith et al., 2013*; *van Kerkoerle et al., 2014*). While *in vivo* laminar data from primate sensorimotor cortex are lacking, *in vitro* recordings from somatosensory and motor cortices demonstrate that beta activity is generated in neural circuits dominated by infragranular layer V pyramidal cells (*Roopun et al., 2006*; *Roopun et al., 2010*; *Yamawaki et al., 2008*). By contrast, gamma activity is thought to arise from supragranular layers II/III of mouse somatosensory cortex (*Cardin et al., 2009*; *Carlén et al., 2012*). The results presented here support generalized theories of laminar organization across cortex, and are the first to non-invasively provide evidence for the laminar origin of movement-related sensorimotor activity.

## High frequency activity in visual cortex is enhanced by mismatches between possible feedforward and feedback signals

We found that visual gamma was enhanced following the presentation of the instruction cue in incongruent compared to congruent trials. This was in agreement with our predictions, based on the fact that supragranular layer gamma activity is implicated in feedforward processing (*van Kerkoerle et al., 2014*). In our task, the direction of coherent motion in the RDK was congruent with the direction of the following instruction cue in most trials. Participants could therefore form a sensory expectation of the direction of the forthcoming instruction cue, which was violated in incongruent trials. The enhancement of visual gamma following incongruent cues is therefore consistent with the gamma activity increase observed in sensory areas during perceptual expectation violations (*Gurtubay et al., 2001*; *Arnal et al., 2011*; *Todorovic et al., 2011*) as well as layer-specific synaptic currents in supragranular cortical layers during performance error processing (*Sajad et al., 2017*).

## Low frequency activity in sensorimotor cortex reflects a combination of potential feedforward and feedback processes

There are numerous theories for the computational role of beta activity in motor systems. Decreases in beta power prior to the onset of a movement predict the selected action (*Donner et al., 2009*; *Haegens et al., 2011*; *de Lange et al., 2013*), whereas the beta rebound following a movement is attenuated by both perturbation-induced movement errors and target errors induced by goal displacement (*Tan et al., 2014*; *Tan et al., 2016*; *Torrecillos et al., 2015*). Our results unify both of these accounts, showing that the level of beta decrease prior to a movement is modulated by the accumulation of sensory evidence predicting the cued movement, while the beta rebound is diminished when the prepared action must be suppressed in order to correctly perform the cued action (corresponding to a shift in reach target used by *Torrecillos et al., 2015*).

While our results cannot directly distinguish between feedback and feedforward processes because we did not assess interactions between brain regions (*Bastos et al., 2015*; *Michalareas et al., 2016*), they suggest that in the sensorimotor system, low frequency activity can reflect both bottom-up and top-down processes depending on the task epoch. This may occur via bottom-up, feedforward projections from intraparietal regions to motor regions (*Platt and Glimcher, 1999*; *Hanks et al., 2006*; *Tosoni et al., 2008*; *Kayser et al., 2010*) or top-down, feedback projections from the dorsolateral prefrontal cortex (*Heekeren et al., 2004*; *Heekeren et al., 2006*; *Curtis and Lee, 2010*; *Hussar and Pasternak, 2013*; *Georgiev et al., 2016*). The dissociation between bottom-up and top-down influences during different task epochs could indicate that the decrease in beta and the following rebound are the result of functionally distinct processes.

## Future directions

Our ROI-based comparison of deep and superficial laminae can only determine the origin of the strongest source of activity, which does not imply that activity within a frequency band is exclusively confined to either deep or superficial sources within the same patch of cortex (*Maier et al., 2010*; *Bollimunta et al., 2011*; *Spaak et al., 2012*; *Xing et al., 2012*; *Smith et al., 2013*; *Haegens et al., 2015*). We should also note that in all of our control studies, in which we discard spatial information, a bias towards the superficial (pial) cortical surface was present. However, this bias does not increase with SNR for high frequency activity with poor anatomical models (*Figure 5—figure supplement 4*), mirroring the results of simulations showing that this type of laminar analysis is biased superficially at low SNR levels (*Bonaiuto et al., 2018*). Moreover, we used white matter and pial surface meshes to represent deep and superficial cortical laminae, respectively, and therefore made no attempt to

explicitly account for activity arising from the granular layers. Recent studies have shown that beta, and perhaps gamma, activity is generated by stereotyped patterns of proximal and distal inputs to infragranular and supragranular pyramidal cells (*Lee and Jones, 2013*; *Jones, 2016*; *Sherman et al., 2016*).

Finally, a new generation of wearable MEG sensors, optically pumped magnetometers (*Boto et al., 2016*; *Boto et al., 2017*; *Boto et al., 2018*), promises to extend the reach of laminar MEG. These sensors do not require cryogenic cooling and can therefore be placed directly on the scalp surface, directly increasing SNR. This allows participants to make relatively unconstrained and natural movements; future such systems, which were comfortable to wear, would give the possibility of further augmenting the SNR by recording over much longer periods (*Boto et al., 2018*). Such flexibility in participant behavior opens the door to the possibility of testing theories about the changes in hierarchical communication in the brain, either developmentally or in patient populations such as those with movement disorders, autism spectrum disorders and schizophrenia (*Wang, 2010*; *Wilson et al., 2011*; *Gandal et al., 2012*; *Wright et al., 2012*; *Chan et al., 2016*; *Kessler et al., 2016*; *Liddle et al., 2016*).

## Materials and methods

### Behavioral task

Eight neurologically healthy volunteers participated in the experiment (six male, aged 28.5 ± 8.52 years). The study protocol was in full accordance with the Declaration of Helsinki, and all participants gave written informed consent after being fully informed about the purpose of the study. The study protocol, participant information, and form of consent, were approved by the UCL Research Ethics Committee (reference number 5833/001). Participants completed a visually cued action decision making task in which they responded to visual stimuli projected on a screen by pressing one of two buttons on a button box using the index and middle finger of their right hand. On each trial, participants were required to fixate on a small (0.5°×0.5°) white cross in the center of a screen. After a baseline period randomly varied between 1 s and 2 s, a random dot kinematogram (RDK) was displayed for 2 s with coherent motion either to the left or to the right (*Figure 1A*). Following a 500 ms delay, an instruction cue appeared, consisting of a 3°×1° arrow pointing either to the left or the right, and participants were instructed to press the corresponding button (left or right) as quickly and as accurately as possible. Trials ended once a response had been made or after a maximum of 1 s if no response was made.

The task had a factorial design with congruence (whether or not the direction of the instruction cue matched that of the coherent motion in the RDK) and coherence (the percentage of coherently moving dots in the RDK) as factors (*Figure 1B*). Participants were instructed that in most of the trials (70%), the direction of coherent motion in the RDK was congruent to the direction of the instruction cue. Participants could therefore reduce their mean response time (RT) by preparing to press the button corresponding to the direction of the coherent motion. The RDK consisted of a 10°×10° square aperture centered on the fixation point with 100, 0.3° diameter dots, each moving at 4°/s. On each trial, a certain percentage of the dots (specified by the motion coherence level) moved coherently through the aperture in one direction, left or right. The remaining dots moved in random directions through the aperture, with a consistent path per dot. The levels were individually set for each participant by using an adaptive staircase procedure (QUEST; *Watson and Pelli, 1983*) to determine the motion coherence at which they achieved 82% accuracy in a block of 40 trials at the beginning of each session, in which they had to simply respond with the left or right button to leftwards or rightwards motion coherence. The resulting level of coherence was then used as medium, and 50% and 150% of it as low and high, respectively.

Each block contained 126 congruent trials, and 54 incongruent trials, and 60 trials for each coherence level with half containing coherent leftward motion, and half rightward (180 trials total). All trials were randomly ordered. Participants completed three blocks per session, and 1–5 sessions on different days, resulting in 540–2700 trials per participant (M = 1822.5, SD = 813.21). The behavioral task was implemented in MATLAB (The MathWorks, Inc., Natick, MA) using the Cogent 2000 toolbox (http://www.vislab.ucl.ac.uk/cogent.php).

## MRI acquisition

Prior to MEG sessions, participants underwent two MRI scanning protocols during the same visit: one for the scan required to generate the scalp image for the head-cast, and a second for MEG source localization. Structural MRI data were acquired using a 3T Magnetom TIM Trio MRI scanner (Siemens Healthcare, Erlangen, Germany), while participants were laying in a supine position.

The first protocol was used to generate an accurate image of the scalp for head-cast construction (*Meyer et al., 2017a*). This used a T1-weighted 3D spoiled fast low angle shot (FLASH) sequence with the following acquisition parameters: 1 mm isotropic image resolution, field-of view set to 256, 256, and 192 mm along the phase (anterior-posterior, A–P), read (head-foot, H–F), and partition (right-left, R–L) directions, respectively. The repetition time was 7.96 ms and the excitation flip angle was 12°. After each excitation, a single echo was acquired to yield a single anatomical image. A high readout bandwidth (425 Hz/pixel) was used to preserve brain morphology and no significant geometric distortions were observed in the images. Acquisition time was 3 min 42 s, a sufficiently short time to minimize sensitivity to head motion and any resultant distortion. Care was also taken to prevent distortions in the image due to skin displacement on the face, head, or neck, as any such errors could compromise the fit of the head-cast. Accordingly, a more spacious 12 channel head coil was used for signal reception without using either padding or headphones.

The second protocol was a quantitative multiple parameter mapping (MPM) protocol, consisting of 3 differentially-weighted, RF and gradient spoiled, multi-echo 3D FLASH acquisitions acquired with whole-brain coverage at 800 µm isotropic resolution. Additional calibration data were also acquired as part of this protocol to correct for inhomogeneities in the RF transmit field (*Lutti et al., 2010*; *Lutti et al., 2012*; *Callaghan et al., 2015*). For this protocol, data were acquired with a 32-channel head coil to increase SNR.

The FLASH acquisitions had predominantly proton density (PD), T1 or magnetization transfer (MT) weighting. The flip angle was 6° for the PD- and MT-weighted volumes and 21° for the T1 weighted acquisition. MT-weighting was achieved through the application of a Gaussian RF pulse 2 kHz off resonance with 4 ms duration and a nominal flip angle of 220° prior to each excitation. The field of view was set to 224, 256, and 179 mm along the phase (A–P), read (H–F), and partition (R–L) directions, respectively. Gradient echoes were acquired with alternating readout gradient polarity at eight equidistant echo times ranging from 2.34 to 18.44 ms in steps of 2.30 ms using a readout bandwidth of 488 Hz/pixel. Only six echoes were acquired for the MT-weighted acquisition in order to maintain a repetition time (TR) of 25 ms for all FLASH volumes. To accelerate the data acquisition and maintain a feasible scan time, partially parallel imaging using the GRAPPA algorithm (*Griswold et al., 2002*) was employed with a speed-up factor of 2 and forty integrated reference lines in each phase-encoded direction (A-P and R-L).

To maximize the accuracy of the measurements, inhomogeneity in the transmit field was mapped by acquiring spin echoes and stimulated echoes across a range of nominal flip angles following the approach described in *Lutti et al., 2010*, including correcting for geometric distortions of the EPI data due to B0 field inhomogeneity. Total acquisition time for all MRI scans was less than 30 min.

Quantitative maps of proton density (PD), longitudinal relaxation rate (R1 = 1/T1), magnetization transfer saturation (MT) and effective transverse relaxation rate (R2*=1/T2*) were subsequently calculated according to the procedure described in *Weiskopf et al. (2013)*. Each quantitative map was co-registered to the scan used to design the head-cast, using the T1 weighted map. The resulting maps were used to extract cortical surface meshes using FreeSurfer (see below).

## Head-cast construction

From an MRI-extracted image of the skull, a head-cast that fit between the participant's scalp and the MEG dewar was constructed (*Troebinger et al., 2014b*; *Meyer et al., 2017a*). Scalp surfaces were first extracted from the T1-weighted MRI scans acquired in the first MRI protocol using standard SPM12 procedures (RRID:SCR_007037; http://www.fil.ion.ucl.ac.uk/spm/). Next, this tessellated surface was converted into the standard template library (STL) format, commonly used for 3D printing. Importantly, this conversion imposed only a rigid body transformation, meaning that it was easily reverse-transformable at any point in space back into native MRI space. Accordingly, when the fiducial locations were optimized and specified in STL space as coil-shaped protrusions on the scalp, their exact locations could be retrieved and employed for co-registration. Next, the head-cast

design was optimized by accounting for factors such as head-cast coverage in front of the ears, or angle of the bridge of the nose. To specify the shape of the fiducial coils, a single coil was 3D scanned and three virtual copies of it were placed at the approximate nasion, left peri-auricular (LPA), and right peri-auricular (RPA) sites, with the constraint that coil placements had to have the coil-body and wire flush against the scalp, in order to prevent movement of the coil when the head-cast was worn. The virtual 3D model was placed inside a virtual version of the scanner dewar such that the distance to the sensors was minimized (by placing the head as far up within the dewar as possible) while ensuring that vision was not obstructed. Next, the head-model (plus spacing elements and coil protrusions) was printed using a Zcorp 3D printer (Zprinter 510) with 600 × 540 dots per inch resolution. The 3D printed head model was then placed inside the manufacturer-provided replica of the dewar and liquid resin was poured in between the surfaces to fill the negative space, resulting in the participant-specific head-cast. The fiducial coil protrusions in the 3D model now become indentations in the resulting head-cast, in which the fiducial coils can sit during scanning. The anatomical landmarks used for determining the spatial relationship between the brain and MEG sensors are thus in the same location for repeated scans, allowing data from multiple sessions to be combined (*Meyer et al., 2017a*).

## FreeSurfer surface extraction

FreeSurfer (v5.3.0; *Fischl, 2012*) was used to extract cortical surfaces from the multi-parameter maps. Use of multi-parameter maps as input to FreeSurfer can lead to localized tissue segmentation failures due to boundaries between the pial surface, dura mater and CSF showing different contrasts compared to that assumed within FreeSurfer algorithms (*Lutti et al., 2014*). Therefore, an in-house FreeSurfer surface reconstruction procedure was used to overcome these issues, using the PD and T1 maps as inputs. Detailed methods for cortical surface reconstruction can be found in *Carey et al., 2017*. This process yields surface extractions for the pial surface (the most superficial layer of the cortex adjacent to the cerebro-spinal fluid, CSF), and the white/grey matter boundary (the deepest cortical layer). Each of these surfaces is downsampled by a factor of 10, resulting in two meshes comprising about 30,000 vertices each (M = 30,0940.75, SD = 2,665.450.45 over participants). For the purposes of this study, we used these two surfaces to represent deep (white/grey interface) and superficial (grey-CSF interface) cortical models.

## MEG acquisition

MEG recordings were made using a 275-channel Canadian Thin Films (CTF) MEG system with superconducting quantum interference device (SQUID)-based axial gradiometers (VSM MedTech, Vancouver, Canada) in a magnetically shielded room. The data collected were digitized continuously at a sampling rate of 1200 Hz. A projector displayed the visual stimuli on a screen (~50 cm from the participant), and participants made responses with a button box. All data are archived at the Open MEG Archive (OMEGA; *Niso et al., 2016*) and may be accessed via http://dx.doi.org/10.23686/0015896 (*Niso et al., 2018*).

## Behavioral analyses

Participant responses were classified as correct when the button pressed matched the direction of the instruction cue, and incorrect otherwise. The response time (RT) was measured as the time of button press relative to the onset of the instruction cue. We analyzed accuracy using a generalized linear mixed model with a logit link function, using correct (true or false) in each trial as the dependent variable, congruence (congruent or incongruent) and coherence (low, medium, high) and their interaction as fixed effects, and a participant-specific intercept as a random effect. Fixed effects were tested using type III Wald $\chi^2$ tests. RT was analyzed using a linear mixed model also using congruence as coherence and their interaction as fixed effects, with a participant-specific intercept as a random effect. Fixed effects for this model were estimated using type III Wald F tests with Kenward-Rogers approximated degrees of freedom (*Kenward and Roger, 1997*). For both models, planned pairwise follow-up tests were performed using LSMEANS between congruence levels at each coherence level, Tukey corrected.

## MEG preprocessing

All MEG data preprocessing and analyses were performed using SPM12 (RRID:SCR_007037; http://www.fil.ion.ucl.ac.uk/spm/) using Matlab R2014a (RRID:SCR_001622) and are available at http://github.com/jbonaiuto/meg-laminar (*Bonaiuto, 2018*; copy archived at https://github.com/elifesciences-publications/meg-laminar). The data were filtered (5th order Butterworth bandpass filter: 2–100 Hz, Notch filter: 50 Hz) and downsampled to 250 Hz. Eye-blink artifacts were removed using multiple source eye correction (*Berg and Scherg, 1994*). Trials were then epoched from 1 s before RDK onset to 1.5 s after instruction cue onset, and from 2 s before the participant's response to 2 s after. Blocks within each session were merged, and trials whose variance exceeded 2.5 standard deviations from the mean were excluded from analysis.

## Reproducibility analysis

The reproducibility of the topographic maps, ERFs, and time frequency decompositions was quantified for a representative participant by computing the intra-class correlation coefficient (ICC), a measure of test-retest reliability (*Shrout and Fleiss, 1979*). This was done within-session over runs used a type 2 k ICC, with the runs modeled as a random effect and the measure given by an average over trials within a run. Similarly, the between-session ICC was type 2 k with sessions modeled as a random effect and the measure given by an average over runs within a session.

## Sensor-level analysis

At the sensor-level, we analyzed three epochs: one aligned to the RDK stimulus (0 – 2000 ms), one centered on instruction stimulus (−500 ms to +500 ms), and one centered on the participant's response (−1000 ms to +1000 ms), with 250 ms padding on either side to avoid edge effects. For each epoch type, seven-cycle Morlet wavelets were used to compute power within 2 – 45 Hz in increments of 1 Hz, and a multi-taper analysis was used to computer power within 55 – 115 Hz in increments of 5 Hz (sine taper, time resolution = 200 ms, time step = 10 ms). Power for each epoch type was baseline-corrected using the 500 ms prior to the onset of the RDK stimulus in a frequency-specific manner using robust averaging. Robust averaging is a form of general linear modeling (*Wager et al., 2005*) used to reduce the influence of outliers on the mean by iteratively computing a weighting factor for each sample according to how far it is from the mean. The baseline-corrected time-frequency spectrograms were then averaged over a cluster of 15 sensors overlying occipital cortex for visual signals (MLO53, MLO43, MLO32, MLO52, MLO31, MLO51, MLO41, MZO02, MZO03, MRO52, MRO42, MRO31, MRO53, MRO43, MRO32) and 18 sensors overlying contralateral motor cortex for sensorimotor signals (MLC17, MLC25, MLC32, MLC42, MLC54, MLC63, MRC63, MLP57, MLP45, MLP35, MLP12, MLP23, MLC55, MZC04, MLP44, MLP34, MLP22, MLP11), and finally smoothed using a Gaussian kernel (FWHM 8 × 8 Hz frequency bins and 80 ms). We used a linear mixed model with subject-specific offsets as random effects to test for significant changes in power from baseline. We used a significance threshold of p<0.05, Bonferroni corrected for multiple comparisons (over time and frequency).

## Source reconstruction

Source inversion was performed using the empirical Bayesian beamformer (EBB; *Belardinelli et al., 2012*; *López et al., 2014*). The sensor data were first reduced, using singular value decomposition to 180 virtual channels, each with 16 temporal samples (weighting the dominant modes of temporal variation across the window). For uninformative priors, the maximum-likelihood solution to the inverse problem reduces to:

$$\hat{J} = QL^T(Q_\epsilon + LQL^T)^{-1}Y$$

where $\hat{J}$ is the estimated current density across the source space, $Y$ is (reduced) measured data, $L$ is the lead field or sensitivity matrix that can be computed based on the sensor and volume conductor geometry. $Q_\epsilon$ is the sensor noise, and $Q$ is the prior estimate of source covariance. We assumed the sensor level covariance($Q_\epsilon$) to be an identity matrix (see discussion). Most popular inversion algorithms can be differentiated by the form of $Q$ (*Friston et al., 2008*; *López et al., 2014*). Here we used a beamformer prior to estimate the structure of $Q$ (*Belardinelli et al., 2012*; *López et al.,*

*2014*) where a direct estimate of prior source co-variance ($Q$) is made based on the sensor-level data:

$$Q(i) = \frac{1}{L_i^T L_i}(L_i^T (YY^T)^{-1} L_i + \lambda I)^{-1}$$

$Q$ is a diagonal matrix, and each element of the diagonal $Q(i)$ corresponds to a source location $i$. The (reduced) sensor level data is $Y$, the lead field of each element $i$ is $L_i$, $^T$ denotes the transpose operator, $I$ is an identity matrix, and $\lambda$ is a regularization constant. The highest resolution beamformer estimate will be made with $\lambda = 0$ and this is the default used throughout the paper. Such low values of regularization can, however, become problematic especially when comparing signals occupying different bandwidths at different SNRs (*Brookes et al., 2008*). In order to verify that the differential effects we were observed were not due to regularization, we therefore also implemented an augmented EBB solution in which the Bayesian scheme optimized from a range of source priors each at different levels of regularization (0, 5, 10, 50, 100 and 1000) percent of the mean eigenvalue of $YY^T$ (*Figure 5—figure supplement 13B*).

The prior estimates of $Q_\epsilon$ and $Q$ are then re-scaled or optimally mixed using an expectation maximization scheme (*Friston et al., 2008*) to give an estimate of $J$ that maximizes model evidence. The source level prior was based on the beamformer power estimate across a two-layer manifold comprised of pial and white cortical surfaces with source orientations defined as normal to the cortical surface and a spatial coherence prior (*Friston et al., 2008*), $G(\sigma) : \sigma = 0.4$ (corresponding to a FWHM of approximately 4 mm). We used the Nolte single shell head model (*Nolte, 2003*). All analyses were carried out using the SPM12 (RRID:SCR_007037; http://www.fil.ion.ucl.ac.uk/spm/) software package (see *López et al., 2014*) for implementation details).

## Analyses for laminar discrimination

The laminar analysis reconstructed the data onto a mesh combining the pial and white matter surfaces, thus providing an estimate of source activity on both surfaces (*Figure 4*). We analyzed six different visual and sensorimotor signals at different frequencies and time windows of interest (WOIs), using the same frequency bands across participants: RDK-aligned visual alpha (7-13Hz; WOI=[0s, 2s]; baseline WOI=[-1s, -.5s]), RDK-aligned visual gamma (60-90Hz; WOI=[250ms, 500ms]; baseline WOI=[-500ms, -250ms]), instruction cue-aligned visual gamma (60-90Hz; WOI=[100ms, 500ms]; baseline WOI=[-500ms, -100ms]), RDK-aligned sensorimotor beta (15-30Hz; WOI=[0s, 2s]; baseline WOI= [-500ms, 0ms]), response-aligned sensorimotor beta (15-30Hz; WOI=[500ms, 1s]; baseline WOI=[- 250ms 250ms]), and response-aligned sensorimotor gamma (60-90Hz; WOI=[-100ms, 200ms]; baseline WOI=[-1.5s, -1s]). For each signal, we defined an ROI by comparing power in the associated frequency band during the WOI with a prior baseline WOI at each vertex and averaging over trials. Vertices in either surface with a mean unsigned fractional change in power from the baseline in the 80[th] percentile over all vertices on that surface (the top 20%), as well as the corresponding vertices on the other surface, were included in the ROI. This ensured that the contrast used to define the ROI was orthogonal to the subsequent pial versus white matter surface contrast. For each trial, ROI values for the pial and white matter surfaces were computed by averaging the unsigned fractional change in power compared to baseline in that surface within the ROI ($\frac{|WOI - baseline|}{baseline}$).

For within-participant tests, a paired t-test was used to compare the ROI values from the pial surface with those from the white matter surface over trials (*Figure 4*). This resulted in positive t-statistics when the unsigned fractional change in power from baseline was greatest on the pial surface, and negative values when the fractional change was greatest on the white matter surface. All t-tests were performed with corrected noise variance estimates in order to attenuate artifactually high significance values (*Ridgway et al., 2012*). Group-level statistics were performed using one-sample Wilcoxon tests of the unsigned fractional change in power from baseline averaged within ROI ($\frac{|WOI_{pial} - baseline_{pial}|}{baseline_{pial}} - \frac{|WOI_{whitematter} - baseline_{whitematter}|}{baseline_{whitematter}}$).

The control analyses utilized the same procedure, but each introduced some perturbation to the data. The shuffled analysis permuted the lead fields of the forward model prior to source reconstruction in order to destroy any correspondence between the cortical surface geometry and the sensor data. This was repeated 10 times per session, with a different random lead field permutation each time. The mean unsigned magnitude of the change in power from baseline averaged within ROI was

then used as the null hypothesis in the follow-up runs of the main laminar analyses. Each permutation was then used in the laminar analysis for every low and high frequency signal. The co-registration error analysis introduced a rotation (M = 10°, SD = 2.5°) and translation (M = 10 mm, SD = 2.5 mm) of the fiducial coil locations in a random direction prior to source inversion, simulating between-session co-registration error. This was done 10 times per session, with a different random rotation and translation each time. Again, each perturbation was used in the laminar analysis for every low and high frequency signal. The SNR analysis used a random subset of the available trials from each participant, gradually increasing the number of trials used from 10 to the number of trials available. This was repeated 10 times, using a different random subset of trials each time, and the resulting t-statistics were averaged. The white noise analysis was used to decrease SNR by progressively adding Gaussian white noise of increasing standard deviation to the sensor level data.

For analyses of laminar bias, distance to the scalp was computed using the CAT12 toolbox (http://dbm.neuro.uni-jena.de/cat/) to generate a convex hull surface from the pial surface, and then computing the Euclidean distance between each vertex and the nearest vertex on hull surface (*Van Essen, 2005*; *Im et al., 2006*; *Tosun et al., 2015*), and lead field strength was computed as root mean square of the lead field. Relationships between relative lead field strength or laminar depth and the effect size of the laminar bias were evaluated using per-participant Spearman partial correlation coefficients (controlling for the effect of laminar depth or relative lead field strength, respectively). Each participant's correlation coefficient was Fisher-transformed and the resulting Z scores were compared against zero using a one sample t-test. The analysis using only vertices where the white matter is closer to the scalp used the same ROIs as the main analysis, but only including vertex pairs where the sulcal depth of the white matter vertex was less than that of the pial vertex. The analysis controlling for the effect of the distance to the scalp used robust regression (*Holland and Welsch, 1977*) to fit a linear model to the difference (pial – white matter) of the unsigned fractional change in power from baseline, averaged over trials. The square root of the distance to scalp surface (averaged over pial and white matter vertices within each vertex pair) was used as the independent variable. The main laminar analysis was then run on the residuals of this regression.

The patch size analysis ran each inversion using a range of reconstruction patch sizes (FWHM = 2.5, 5, 10, and 20 mm), and compared the free energy metric of model fit of each to the mean over all patch sizes.

## Condition comparison

For each visual and sensorimotor frequency band/task epoch combination, induced activity was compared between task conditions on the surface and within the anatomically constrained ROI identified from the corresponding laminar analysis. Seven-cycle Morlet wavelets were used to compute power within the frequency band and this was baseline-corrected in a frequency-specific manner using robust averaging. For each participant, the mean percent change in power over the WOI was averaged over all trials within every condition. Wilcoxon tests for comparing two repeated measures were used to compare the change in power for instruction cue-aligned visual gamma and sensorimotor beta rebound between congruent and incongruent trials. A Friedman test for comparing multiple levels of a single factor with repeated measures was used to compare the sensorimotor beta decrease between low, medium, and high RDK coherence trials. This was followed up by Tukey-Kramer corrected pairwise comparisons. Only trials in which a correct response was made were analyzed.

## Acknowledgements

JB and HR were supported by a BBSRC research grant (BB/M009645/1). SM was supported by a Medical Research Council and Engineering and Physical Sciences Research Council grant MR/K6010/86010/1, the Medical Research Council UKMEG Partnership grant MR/K005464/1, and a Wellcome Principal Research Fellowship to Neil Burgess. SL was supported by a Wellcome Trust clinical post-doctoral grant (105804/Z/14/Z). The Wellcome Centre for Human Neuroimaging is supported by core funding from the Wellcome (203147/Z/16/Z). The funders had no role in the preparation of the manuscript.

## Additional information

### Funding

| Funder | Grant reference number | Author |
|---|---|---|
| Biotechnology and Biological Sciences Research Council | BB/M009645/1 | James J Bonaiuto |
| Medical Research Council | MR/K6010/86010/ | Sofie S Meyer |
| Engineering and Physical Sciences Research Council | MR/K005464/1 | Sofie S Meyer |
| Wellcome Trust | 105804/Z/14/Z | Simon Little |

The funders had no role in the preparation of the manuscript.

### Author contributions

James J Bonaiuto, Conceptualization, Formal analysis, Investigation, Methodology, Writing—original draft, Writing—review and editing; Sofie S Meyer, Investigation, Methodology, Writing—original draft, Writing—review and editing; Simon Little, Holly Rossiter, Martina F Callaghan, Frederic Dick, Methodology, Writing—original draft, Writing—review and editing; Gareth R Barnes, Sven Bestmann, Conceptualization, Supervision, Funding acquisition, Methodology, Writing—original draft, Writing—review and editing

### Author ORCIDs

James J Bonaiuto http://orcid.org/0000-0001-9165-4082
Simon Little http://orcid.org/0000-0001-6249-6230
Frederic Dick http://orcid.org/0000-0002-2933-3912
Sven Bestmann http://orcid.org/0000-0002-6867-9545

### Ethics

Human subjects: The study protocol was in full accordance with the Declaration of Helsinki, and all participants gave written informed consent after being fully informed about the purpose of the study. The study protocol, participant information, and form of consent, were approved by the UCL Research Ethics Committee (reference number 5833/001).

### Decision letter and Author response

Decision letter https://doi.org/10.7554/eLife.33977.039
Author response https://doi.org/10.7554/eLife.33977.040

## Additional files

### Supplementary files

• Transparent reporting form
DOI: https://doi.org/10.7554/eLife.33977.033

### Data availability

All data are archived at the Open MEG Archive (OMEGA; http://dx.doi.org/10.23686/0015896). All data analysis code is available on github (https://github.com/jbonaiuto/meg-laminar; copy archived at https://github.com/elifesciences-publications/meg-laminar). Numerical data for figures 1-3, 5-8 are included as source data files.

The following dataset was generated:

| Author(s) | Year | Dataset title | Dataset URL | Database, license, and accessibility information |
|---|---|---|---|---|
| Niso G., Rogers C., | 2018 | The Open MEG Archive (OMEGA) | http://dx.doi.org/10. | Publicly available at |

Moreau JT., Chen L-Y., Madjar C., Das S., Bock E., Tadel F., Evans A. C., Jolicoeur P., Baillet S.

23686/0015896

the Open MEG Archive (OMEGA).

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
