## [Decision Letter]

Thank you for submitting your article "Laminar-specific cortical dynamics in human visual and sensorimotor cortices" for consideration by *eLife*. Your article has been reviewed by three peer reviewers, including Tobias H Donner as the Reviewing Editor, and the evaluation has been overseen by Richard Ivry as the Senior Editor.

The reviewers have now discussed the reviews with one another and the Reviewing Editor has drafted this decision to help you prepare a revised submission.

Summary:

This manuscript investigates brain rhythms in humans and focuses on laminar differences. Several studies in non-human primates have shown that the cortical layers differ with regard to the strength of alpha-beta and gamma rhythms, respectively. Also, inter-areal Granger causality in those bands is related to inter-areal connections with their typical laminar projection pattern, and this has been found also in human subjects. Some of the authors have recently developed a technique for stabilizing subjects' heads in the MEG. In the current manuscript, this technique is used to investigate whether the laminar difference in the strengths of alpha-beta and gamma rhythms can be found in human subjects. MEG data of several subjects is obtained, while they perform a visual or sensorimotor task. A generative model is constructed for each subject based on a surface mesh including both the white matter and pial surfaces. Indeed, the results show that alpha-beta is stronger on the white-matter mesh and gamma stronger on the pial mesh.

Essential revisions:

All reviewers agreed that your study addresses an important and timely issue in systems neuroscience. They also agreed that your methods are overall strong and innovative, and that your findings are interesting. That said, your goal is very ambitious, and all reviewers felt that further important control analyses were necessary to make your claims fully convincing.

1) Further steps to address the SNR confound.

The most important concern shared by all reviewers is the potential SNR confound: The white-matter mesh is on average more distant from the head surface than the pial mesh, and source estimation can be influenced by this distance to the head surface. Laminar MEG analysis is biased toward superficial laminae as a function of SNR (Bonaiuto, 2018), and SNR differs between different rhythms. You partially address this issue by showing similar laminar specificity for beta decreases and beta rebounds. However, this does not fully remove concerns about differences between different rhythms because power is generally larger in the beta-band than in the gamma-band.

The reviewers converged on the following three points for addressing this concern.

1a) An analysis of pairs of vertices, with each pair containing a pial and a corresponding white-matter vertex.

Across all those pairs, the white-matter vertex is almost certainly more distant from the head surface than the pial vertex. However, there are many pairs, for which the white-matter vertex is closer to the head surface than the pial vertex. This can be seen e.g. in Figure 3A and Figure 4—figure supplement 1B. You should test, whether also for those pairs, the laminar preferences of alpha-beta and of gamma hold.

1b) A test of whether laminar preferences are correlated with distances from head surface. You could e.g. derive for each vertex pair two laminar indices, one for power and another for distance from head surface, and then correlate those indices. Ideally, there should be no such correlation, because there is no reason to assume that the physiological specificity of rhythms for layers is related to whether the respective cortical patch is on the surface, or in a sulcus with a reversed distance-to-surface relationship. In fact, Buffalo et al., 2011 have shown that the laminar specificity for rhythms holds both on the surface (V1, V4), and in the depth (V2), where laminar distance from the head surface is reversed.

1c) Showing the full cortical distribution of the laminar bias estimates.

Please show plots like Figure 3B bottom, but for the data in Figures 4 and 5. The laminar-bias (pial-whit matter t-score) is the key measure for the present study, which is assessed all over the cortex. The result of this analysis should be shown. Relatedly, you should show full slices of beamforming on a regular, high-resolution grid. This will help assess the true resolution and spatial structure of the source-estimate, including potential laminar effects.

2) Elaborate on the generative inverse solution (EBB) used here. This technique is obviously crucial for assessing the validity of the present findings. Reviewers felt the explanation needs to be clarified. For example, what are the temporal modes in the subsection “Source reconstruction”? Also, the rest of the description (– hyper-parameters? covariance priors? why can the sensor level covariance be assumed to be an identity matrix? etc.) requires more detail. When expanding this description, please do this with the following question in mind: Can any of these points affect the laminar estimates differentially for the different frequency bands?

3) Discuss the patch-size confound that was shown in Bonaiuto et al., 2018.

The reported laminar bias may reflect the patch-size dependent laminar bias of the employed method. In their previous publication (Bonaito et al., 2018), you did not only demonstrate a SNR-bias, but also a patch-size bias. Laminar estimates were biased towards superficial and deep sources depending on the mismatch between the size of estimated patches and the local dispersion of current flow. The local correlation structure differs between alpha and gamma-band activity, which in turn may induce a laminar bias. This potential problem should be discussed.

4) Present group-level statistics throughout the paper. The manuscript presents t-statistics separately for each of 8 subjects. You should perform additional tests, in which subjects are combined. For some effects, this is done, e.g. in the last paragraph of the subsection “Deep sensorimotor beta scales with RDK motion coherence and cue congruence”, however, it should be done throughout.

5) Further characterize time-frequency representations (TFRs) of visual stimulus responses. TFRs of the group average power modulations (sensor level) should be shown before proceeding to the laminar results (Figures 3-5). You only show the TFRs from a single participant (Figure 2), and in this participant, the gamma-band responses to RDKs and cue are not so compelling. Especially the responses to the sustained RDK presentation do not seem to replicate the sustained stimulus-induced gamma-band responses from earlier MEG work. Do the group average TFRs show stronger visual gamma-band responses? Are those statistically significant (within and/or across participants)?

6) Please change the color scale in Figure 4.

The color maps in Figure 4 are asymmetrically compressed such that effects in the reported and unreported direction are mapped more onto colorful and black colors, respectively. This might bias appearance of the data towards the reported effects.

[Editors' note: further revisions were requested prior to acceptance, as described below.]

Thank you for submitting your article "Lamina-specific cortical dynamics in human visual and sensorimotor cortices" for consideration by *eLife*. Your article has been reviewed by two peer reviewers, and the evaluation has been overseen by Tobias Donner as the Reviewing Editor and Richard Ivry as the Senior Editor. The reviewers have opted to remain anonymous.

The reviewers have discussed the reviews with the Reviewing Editor, who has drafted this decision to help you prepare a revised submission.

Summary:

We have provided a summary of your paper in the decision letter for the initial submission. Both reviewers and the Reviewing Editor were impressed by your thorough revision of the paper and your detailed replies to the reviewers' comments. You have addressed all of their previously raised concerns, which has substantially improved the paper. We are happy to inform you that we would be happy to publish your paper at *eLife*, in principle, provided that you will address some outstanding issues detailed below. In general, both reviewers felt that some of your statements and conclusions would need to be toned down, to more accurately reflect the results of the new control analyses. This is particularly important, given the strong impact that your paper may have on the community. In addition, reviewer #3 thought that your control analyses exposed a few important issues that would need to be discussed.

Reviewer #3 also suggested some additional control analyses that might help rule out these concerns. We leave it to you to decide whether or not to include these analyses in your revision.

Essential revisions:

1) One major concern is that the reported superficial bias for gamma might reflect the dependency of the estimated depth-bias on relative leadfield-strength (pial vs. white matter), in combination with the higher proportion of stronger pial lead-fields and a low SNR for the gamma-band.

Figure 5—figure supplements 5, 6, 7 and 8 are important in this respect. The new result that the alpha/beta depth-bias does not invert for pairs with stronger white-matter lead-fields is important and re-assuring. This corresponds to the negative offset of the regression fits in Figure 5—figure supplement 6B for alpha/beta. This finding provides compelling evidence for a general deep source of alpha/beta.

However, the situation is different for gamma. First, there seems to be hardly any visual gamma response to the RDK stimuli to begin with (see point 2). Second, the regressions in Figure 5—figure supplement 6B (and corresponding correlations, Figure 5—figure supplement 5A) run through the origin (0,0). Thus, there is no general superficial bias for gamma (no offset) if lead-field strength is taken into account. For stronger white-matter lead-fields, the depth bias reverses. Indeed, sensorimotor gamma shows this reversal effect with a deep gamma bias for vertex pairs with stronger white-matter lead-fields (Figure 5—figure supplement 8). The fact that the depth bias does not reverse for vertex pairs with stronger white-matter lead-fields for all other gamma effects does not provide evidence against this interpretation because the number of vertices and the lead-field bias is much lower for pairs with stronger white-matter lead-fields. Together, this suggests, that the general superficial bias of gamma does – at least to a large extent – reflect a confounding effect of the higher proportion of pairs with stronger pial lead-fields, rather than a genuine superficial bias. It is not clear how, then, you can argue for a superficial gamma bias without a significant offset in Figure 5—figure supplement 6B. (You can argue that gamma is generally "more superficial" than alpha/beta.)

In the revision, you should clearly discuss the meaning of (i) the absence of an offset effect for gamma in the correlation analyses (Figure 5—figure supplement 6B), (ii) the biasing effect of a higher proportion of stronger pial lead-fields (marginal in Figure 5—figure supplement 6B), (iii) the reversal of the depth bias for sensorimotor gamma, (iv) the limited statistical inference that can be drawn from an absence of reversals for smaller number of patches.

2) There is no clear visual gamma response during RDK presentation in Figure 3, and there are no quantitative results reported in the text. This is not sufficiently reflected in your description of Figure 3, which should be changed. Furthermore, in this figure the significance mask often directly connects negative and positive effects. How is this possible? One would expect non-significant time-frequency ranges between ranges with significant positive and negative effects. This should be clarified.

3) The concern remains that the reported differences between alpha/beta and gamma are related to SNR-differences between frequencies. First, as mentioned before, the trial omission may not be a sufficient control for this, because omitting trials does not change the mix between signal and noise in the raw data. Indeed, the results caused by this procedure seem rather different than the SNR effects that you reported previously based on mixing signals with variable levels of noise (Bonaiuto et al., 2018). In these previous simulations, lowering SNR by adding noise induced a superficial bias, which does not seem to be the case for the present procedure. You should explicitly mention and discuss the difference and potential drawback of the trial-omission procedure in comparison to the previously applied method.

4) The new figures that show the cortical distribution of the depth bias (Figures 5 and 6) reveal cortical bias-patterns that seem to match the gyral pattern of the brain. In fact, often the depth bias even seems to reverse (superficial vs. deep) depending on the position along the gyri and sulci. This is particularly apparent for all the global effects in Figure 6. Thus, there seems to be a rather strong correlation between the average depth (or lead-field strength) of a vertex pair and its depth bias, which would not be physiologically plausible. This observation is not mentioned. You should discuss it, and comment on whether or not this is problematic.

5) You report that the gamma superficial bias did not increase with SNR for shuffled sensor data (subsection “Sensorimotor beta and gamma originate from distinct cortical laminae”, last paragraph). In contrast, Figure 5—figure supplement 4 shows a stronger superficial bias for high number of trials for gamma and shuffled sensors (gray lines rising towards the right). Do you mean "with lower SNR"? This needs to be clarified.

6) As noted previously, the argument is not valid, that a deep localization of beta during the pre-response suppression and during the beta-rebound is evidence against an SNR confound (subsection “Sensorimotor beta and gamma originate from distinct cortical laminae”, fourth paragraph). Beta is relatively lower during the suppression than during the rebound, but during both intervals absolute beta power is still much larger than absolute gamma power. It is this absolute power level, which is relevant for potential SNR confounds. The argument should be rephrased or removed altogether.

7) You performed a quantification of the cross-session reproducibility. However, it seems that only the results for one subject are reported (subsection “High SNR MEG recordings using individualized head-casts”, last paragraph). The population results should be reported as well (e.g., mean +/- std).

---

## [Author Response]

Essential revisions:All reviewers agreed that your study addresses an important and timely issue in systems neuroscience. They also agreed that your methods are overall strong and innovative, and that your findings are interesting. That said, your goal is very ambitious, and all reviewers felt that further important control analyses were necessary to make your claims fully convincing.1) Further steps to address the SNR confound.The most important concern shared by all reviewers is the potential SNR confound: The white-matter mesh is on average more distant from the head surface than the pial mesh, and source estimation can be influenced by this distance to the head surface. Laminar MEG analysis is biased toward superficial laminae as a function of SNR (Bonaiuto, 2018), and SNR differs between different rhythms. You partially address this issue by showing similar laminar specificity for beta decreases and beta rebounds. However, this does not fully remove concerns about differences between different rhythms because power is generally larger in the beta-band than in the gamma-band.The reviewers converged on the following three points for addressing this concern.1a) An analysis of pairs of vertices, with each pair containing a pial and a corresponding white-matter vertex.Across all those pairs, the white-matter vertex is almost certainly more distant from the head surface than the pial vertex. However, there are many pairs, for which the white-matter vertex is closer to the head surface than the pial vertex. This can be seen e.g. in Figure 3A and Figure 4—figure supplement 1B. You should test, whether also for those pairs, the laminar preferences of alpha-beta and of gamma hold.

We thank the reviewers for this excellent suggestion. Indeed, for a sizable number of pial/white matter vertex pairs, the white matter vertex is closest to the scalp. We have performed the global, functionally defined ROI, and anatomically constrained ROI analyses for each frequency band and task epoch, but now restricted to just those vertex pairs in which the white matter vertex is closest to the scalp.

For ROIs containing only these selected vertices, low and high frequency signals were still separated in terms of their laminar profile (with the exception of motor gamma within the anatomically constrained ROI which was biased toward the white matter surface). Importantly for our main finding in this study, the alpha and beta band signals retained a deep laminae preference. The gamma signals were not biased toward either laminae at the group level. Therefore, our main results were not simply driven by closer proximity of the pial surface to the scalp.

We mention the results from this analysis:

“These analyses were corroborated by analyses showing that … comparing ROIs containing only vertex pairs in which the white matter vertex is closer to the scalp than the pial vertex resulted in a similar pattern of laminar localization (Figure 5—figure supplement 8).”

The results are presented in Figure 5—figure supplement 8.

1b) A test of whether laminar preferences are correlated with distances from head surface. You could e.g. derive for each vertex pair two laminar indices, one for power and another for distance from head surface, and then correlate those indices. Ideally, there should be no such correlation, because there is no reason to assume that the physiological specificity of rhythms for layers is related to whether the respective cortical patch is on the surface, or in a sulcus with a reversed distance-to-surface relationship. In fact, Buffalo et al., 2011 have shown that the laminar specificity for rhythms holds both on the surface (V1, V4), and in the depth (V2), where laminar distance from the head surface is reversed.

Thank you, this is a very useful suggestion. These new plots are now shown in Figure 5—figure supplement 7A. We found no correlation between the relative depth of the two surfaces and our estimate of laminar preference. In Figure 5—figure supplement 7B we also show the analysis for vertices which are matched for depth (+/- 1%). Here we found that the low frequency components consistently (and significantly across the group) localized to the white-matter surface. Although for most participants the high frequency components localized superficially, this was not significant across the group.

However, we were concerned that depth estimates will not account for sensitivity differences due to cortical curvature. Therefore, we conducted a series of additional control analyses, now using the lead field magnitude for each vertex. The lead field magnitude depends on both the distance to the sensors as well as source orientation as determined by cortical curvature.

Figure 5—figure supplement 5A shows that there is a linear relationship between relative lead field strength and our estimate of laminar preference. Signals from all frequency bands are more likely to localize superficially when comparing pial/white matter vertex pairs in which the pial vertex has a stronger lead field. However, across frequency bands, the slope of this relationship remains the same, whereas the offset shifts upwards for high frequencies, reflecting an overall superficial bias (and vice versa for low frequencies). The linear fits to these data are plotted in Figure 5—figure supplement 6; again showing that the slope of the best fit does not change across frequency bands, whereas the offset does.

Finally, we then constrained our analyses to those vertex pairs for which the lead field strengths matched to within 1%. This effectively factors out MEG sensitivity from the comparison. Figure 5—figure supplement 5B shows the results for this analysis. Again we found that the low frequency components consistently (and significantly across the group) localized to the deeper cortical laminae; however the higher frequency signal components exhibited no significant laminar preference at the group level.

We report these results:

“One concern is that our results could have been driven by the relative distance of a given vertex pair from the scalp surface (and hence the MEG sensors). […] These analyses were corroborated by analyses showing that the distance to the scalp surface did not trivially determine laminar preference (Figure 5—figure supplement 7), and that comparing ROIs containing only vertex pairs in which the white matter vertex is closer to the scalp than the pial vertex resulted in a similar pattern of laminar localization (Figure 5—figure supplement 8).

We discuss these results:

“Additionally, we established that our results were not simply caused by the relative strength of the pial and white matter surface lead fields. […] This suggests that our main analyses were sensitive to the likely source of low- and high-frequency signals (rather being simply dependent on the relative magnitude of the influence of source activity from the pial versus white matter surface on the MEG sensors).”

The results are presented in Figure 5—figure supplements 6-7.

1c) Showing the full cortical distribution of the laminar bias estimates.Please show plots like Figure 3B bottom, but for the data in Figures 4 and 5. The laminar-bias (pial-whit matter t-score) is the key measure for the present study, which is assessed all over the cortex. The result of this analysis should be shown.

We have added the proposed plots (now Figures 5 and 6), which show the difference of the unsigned fractional change in power from baseline between the pial and white matter surfaces. The changes are reported in the legend to Figure 5.

Relatedly, you should show full slices of beamforming on a regular, high-resolution grid. This will help assess the true resolution and spatial structure of the source-estimate, including potential laminar effects.

This is a difficult point for us to address. Much of the power of these analyses comes from using the MRI extracted pial and white matter surfaces. These surfaces give not only source location but critically also orientation. This removes a significant unknown and significantly improves the potential spatial resolution (Hillebrand and Barnes, 2003). Therefore, in order to address this point, we used the natural variation in cortical thickness to probe resolution. The figure below shows probability density for pial and white-matter localizations as a function of cortical depth for the different data segments. Also shown is one sample t-test on the (unsigned) magnitude of the difference between the PDFs as a function of surface separation. We have chosen not to include this in the manuscript, because our goal was not to determine the limits of laminar resolution achievable with MEG, but simply to show that over the normal range of cortical thickness, it is possible to localize low and high frequency induced activity to deep and superficial laminae, respectively.

**Author response image 1. respfig1:** Laminar specificity across the range of cortical thickness. (**A**) The y-axis represents the distance between the pial and white matter vertices (cortical thickness). For each visual and sensorimotor frequency band and task epoch (x axis), the probability density function for pial (blue line) and white matter (red line) classifications is shown over the range of cortical thickness. There is a bias toward the white matter surface where the cortex is thick (~4mm and thicker), and toward the pial surface here the cortex is thinner (~3.5mm). However, the overall pial/white matter bias is driven by the frequency band. (**B**) The y-axis represents the t-statistic from a one-way t-test of the difference in the probability density of cortical classification across each frequency band and task epoch (pial-white for gamma, and white-pial for alpha and beta). The ability to discriminate between laminae drops off when the cortical thickness is below 2.5mm.

2) Elaborate on the generative inverse solution (EBB) used here. This technique is obviously crucial for assessing the validity of the present findings. Reviewers felt the explanation needs to be clarified. For example, what are the temporal modes in the subsection “Source reconstruction”? Also, the rest of the description (hyper-parameters? covariance priors? why can the sensor level covariance be assumed to be an identity matrix? etc.) requires more detail. When expanding this description, please do this with the following question in mind: Can any of these points affect the laminar estimates differentially for the different frequency bands?

Thank you this was a very useful exercise. We have now expanded the Materials and methods section outlined below. We did think of a number of factors that could differentially affected the analysis of the frequency bands. We know that errors in the estimate of the sensor level covariance matrix will have a detrimental effect on beamformer estimates (Brookes et al., 2008).

The main differences between bands are SNR, time window length and signal bandwidth. In Brookes et al., 2008 (Equation 33) the covariance matrix error is shown to be proportional to (2*SNR+1)/(2*Time*Bandwidth). We therefore calculated this quantity for the different experimental conditions for a representative participant over four different sessions (Author response image 2).

**Author response image 2. respfig2:** 

In this plot, the bar height represents the mean error across sessions. As these error estimates are comparable and do not distinguish the high and low-frequency bands (i.e. the error for gamma band signals falls in between that of the alpha and beta band signals) we were able to rule out this potential confound.

Our second concern was that the initial EBB covariance prior – which contains an estimate of source power at each vertex – was made using a beamformer with zero regularization. This step could have been differentially affected by the change in SNR across frequency bands. We therefore implemented an augmented version of the algorithm which had available several source EBB source priors, each constructed at a different level of regularization (and therefore of progressively less spatial detail, but also less sensitive to noise).

Figure 5—figure supplement 11B now shows the laminar estimates based on this algorithm (using priors based on regularization levels of 0, 5, 10, 20, 50, 100, and 1000 percent of the mean eigenvalue of the data covariance matrix) relative to the original. The laminar differentiation did not change with this new algorithm, ruling out regularization as a possible confound. However, the evidence for this solution was marginally (but generally not significantly) worse in all cases – suggesting that the added complexity of this model (with additional priors) is not adding enough accuracy to justify its inclusion (and indeed a value of zero was a good initial choice).

Finally, we were concerned that the structure of the sensor covariance matrix might vary across bands. Unfortunately, we did not record these data during our own experiments but were able to find a sensor noise data set recorded independently from our study around the same time period of our data acquisition. In Figure 5—figure supplement 10, we show noise covariance matrices based on data filtered into our main analysis bands and in Figure 5—figure supplement 11C we show that using these matrices, rather than the identity matrix, as the initial sensor level covariance prior does not affect laminar preference.

We have added this issue to the Discussion:

“One possible confound in our analysis is the estimate of sensor noise. […] However, when using a sensor covariance matrix based on empty room noise measurements, the same pattern of laminar preference was observed (Figure 5—figure supplement 11C).”

We have expanded the description of the source inversion algorithm in the Materials and methods section:

“Source inversion was performed using the empirical Bayesian beamformer (EBB; Belardinelli et al., 2012; López et al., 2014). […] We used the Nolte single shell head model (Nolte, 2003). All analyses were carried out using the SPM12 (RRID:SCR_007037; http://www.fil.ion.ucl.ac.uk/spm/) software package (see López et al., 2014 for implementation details).”

3) Discuss the patch-size confound that was shown in Bonaiuto et al., 2018.The reported laminar bias may reflect the patch-size dependent laminar bias of the employed method. In their previous publication (Bonaito et al., 2018), you did not only demonstrate a SNR-bias, but also a patch-size bias. Laminar estimates were biased towards superficial and deep sources depending on the mismatch between the size of estimated patches and the local dispersion of current flow. The local correlation structure differs between alpha and gamma-band activity, which in turn may induce a laminar bias. This potential problem should be discussed.

This is a very good point. We have run control analyses in a representative participant using a range of patch sizes from 2.5-20mm. Indeed, we found that the estimated patch size does bias the laminar estimate, with smaller patch sizes favoring a superficial localization and larger patch sizes favoring deep laminae localization. We followed the recommendation set forth by Bonaiuto et al., 2018, to use a global fit metric (free energy) to compare source inversions performed on the combined pial/white matter surface in order to determine the appropriate patch size. We found that at the optimal patch size for each source, as well as an intermediate patch size of 5mm (close to the approximately 4mm patch size used in the main analysis), low frequency signals reliably localized to deep laminae and high frequency signals localized to superficial laminae.

We have added a discussion of this issue:

“Finally, as discussed previously (Troebinger et al., 2014a; Bonaiuto et al., 2018), over- or under-estimation of source patch sizes can bias laminar results. […] We acknowledge, however, that our models are based on homogeneous Gaussian patches of activity, which therefore may not be realistic.”

4) Present group-level statistics throughout the paper. The manuscript presents t-statistics separately for each of 8 subjects. You should perform additional tests, in which subjects are combined. For some effects, this is done, e.g. in the last paragraph of the subsection “Deep sensorimotor beta scales with RDK motion coherence and cue congruence”, however, it should be done throughout.

We have added group-level statistics to all laminar analyses. These are now reported in the laminar analysis Results sections (subsections “Visual alpha and gamma have distinct laminar specific profiles” and “Sensorimotor beta and gamma originate from distinct cortical laminae”) and described in the Materials and methods section:

*“*Group-level statistics were performed using one-sample Wilcoxon tests of the unsigned fractional change in power from baseline averaged within ROI

(|WOIpial−baselinepial|baselinepial−|WOIwhitematter−baselinewhitematter|baselinewhitematter)

5) Further characterize time-frequency representations (TFRs) of visual stimulus responses. TFRs of the group average power modulations (sensor level) should be shown before proceeding to the laminar results (Figures 3-5). You only show the TFRs from a single participant (Figure 2), and in this participant, the gamma-band responses to RDKs and cue are not so compelling. Especially the responses to the sustained RDK presentation do not seem to replicate the sustained stimulus-induced gamma-band responses from earlier MEG work. Do the group average TFRs show stronger visual gamma-band responses? Are those statistically significant (within and/or across participants)?

Thank you for this suggestion, and we agree that the addition of the group average power modulations at the sensor level is useful. We have therefore now added sensor-level analyses of visual alpha and gamma, as well as sensorimotor beta and gamma:

Task-related changes in low and high frequency activity

To address our main question about the laminar specificity of different frequency channels in human cortex, we *first examined* task-related low- and high-frequency activity from *sensors overlying* visual and sensorimotor cortices. […] At the sensor-level, we indeed observed these classic average power changes, with a significant decrease in beta power (15-30Hz) prior to and during the participant’s response along with a subsequent rebound, and a burst of response-aligned gamma (60-90Hz) activity (Figure 3B; significant time-frequency windows marked, p<0.05, FDR corrected).”

The procedure for this analysis is described as follows:

“Sensor-level analysis

At the sensor-level, we analyzed three epochs: one aligned to the RDK stimulus (0-2000ms), one centered on instruction stimulus (-500ms to +500ms), and one centered on the participant’s response (-1000ms to +1000ms), with 250ms padding on either side to avoid edge effects. […] We used a significance threshold of *p*<0.05 False Discovery Rate (FDR) cluster corrected for multiple comparisons (over time and frequency).”

6) Please change the color scale in Figure 4.The color maps in Figure 4 are asymmetrically compressed such that effects in the reported and unreported direction are mapped more onto colorful and black colors, respectively. This might bias appearance of the data towards the reported effects.

We have changed the color scales in Figures 4 and 5 (now 5 and 6) to be symmetric.

[Editors' note: further revisions were requested prior to acceptance, as described below.]

Essential revisions:1) One major concern is that the reported superficial bias for gamma might reflect the dependency of the estimated depth-bias on relative leadfield-strength (pial vs. white matter), in combination with the higher proportion of stronger pial lead-fields and a low SNR for the gamma-band.Figure 5—figure supplements 5, 6, 7 and 8 are important in this respect. The new result that the alpha/beta depth-bias does not invert for pairs with stronger white-matter lead-fields is important and re-assuring. This corresponds to the negative offset of the regression fits in Figure 5—figure supplement 6B for alpha/beta. This finding provides compelling evidence for a general deep source of alpha/beta.However, the situation is different for gamma. First, there seems to be hardly any visual gamma response to the RDK stimuli to begin with (see point 2).

Thank you. We now show robust visual and sensorimotor gamma responses when accounting for the fact that after the onset of a visual stimulus, gamma occurs in non-timelocked bursts (see response to point 2).

Second, the regressions in Figure 5—figure supplement 6B (and corresponding correlations, Figure 5—figure supplement 5A) run through the origin (0,0). Thus, there is no general superficial bias for gamma (no offset) if lead-field strength is taken into account. For stronger white-matter lead-fields, the depth bias reverses. Indeed, sensorimotor gamma shows this reversal effect with a deep gamma bias for vertex pairs with stronger white-matter lead-fields (Figure 5—figure supplement 8). The fact that the depth bias does not reverse for vertex pairs with stronger white-matter lead-fields for all other gamma effects does not provide evidence against this interpretation because the number of vertices and the lead-field bias is much lower for pairs with stronger white-matter lead-fields. Together, this suggests, that the general superficial bias of gamma does – at least to a large extent – reflect a confounding effect of the higher proportion of pairs with stronger pial lead-fields, rather than a genuine superficial bias. It is not clear how, then, you can argue for a superficial gamma bias without a significant offset in Figure 5—figure supplement 6B. (You can argue that gamma is generally "more superficial" than alpha/beta.)In the revision, you should clearly discuss the meaning of (i) the absence of an offset effect for gamma in the correlation analyses (Figure 5—figure supplement 6B), (ii) the biasing effect of a higher proportion of stronger pial lead-fields (marginal in Figure 5—figure supplement 6B), (iii) the reversal of the depth bias for sensorimotor gamma, (iv) the limited statistical inference that can be drawn from an absence of reversals for smaller number of patches.

Yes, we agree that the superficial lamina-specificity for gamma depends on relative lead field strength. Based on our simulation work, we expected a marginal bias toward the superficial surface, as lead fields from the pial surface are, on average, 12% stronger than those from the white matter surface (Bonaiuto et al., 2018). We have therefore revised our paper accordingly, stating instead that gamma is more superficial than alpha/beta. We now write:

“However, while the slope of the relationship between relative lead field strength and laminar preference was constant across frequency bands, for gamma signals, this regression fit had an offset of approximately zero (Figure 5—figure supplement 7). […] Given these issues, the conservative conclusion would be that visual and sensorimotor gamma localize more superficially than visual alpha and sensorimotor beta.”

2) There is no clear visual gamma response during RDK presentation in Figure 3, and there are no quantitative results reported in the text. This is not sufficiently reflected in your description of Figure 3, which should be changed. Furthermore, in this figure the significance mask often directly connects negative and positive effects. How is this possible? One would expect non-significant time-frequency ranges between ranges with significant positive and negative effects. This should be clarified.

Studies that have found sustained gamma responses during visual stimulus presentation tend to use multi-taper time frequency decomposition (Hoogenboom et al., 2006; Siegel et al., 2007). This is because, aside from a reliable burst of gamma following the onset of the stimulus, gamma occurs in non-timelocked bursts which cancel out during averaging when using wavelet decomposition (Lachaux et al., 2000; Xing et al., 2012). This is why we used a short time window after the stimulus onset for our laminar analysis of gamma power (the source inversion method we used is not based on multitaper estimates of power). In our results, the significance mask that connected positive and negative effects was due to using subject averages, using a relatively lenient correction for multiple comparisons (FDR). We have re-run our sensor-level time frequency analysis using Morlet wavelet decomposition for lower frequencies (<45Hz) and multi-taper decomposition for higher frequencies (>55Hz), and analyzing these data with linear mixed models to test for significant differences from baseline at each time/frequency pixel, with a more conservative correction procedure (Bonferroni). This shows, as intuited by the reviewer, that visual gamma activity does indeed occur throughout the presentation of the visual stimulus.

3) The concern remains that the reported differences between alpha/beta and gamma are related to SNR-differences between frequencies. First, as mentioned before, the trial omission may not be a sufficient control for this, because omitting trials does not change the mix between signal and noise in the raw data. Indeed, the results caused by this procedure seem rather different than the SNR effects that you reported previously based on mixing signals with variable levels of noise (Bonaiuto et al., 2018). In these previous simulations, lowering SNR by adding noise induced a superficial bias, which does not seem to be the case for the present procedure. You should explicitly mention and discuss the difference and potential drawback of the trial-omission procedure in comparison to the previously applied method.

Thanks for raising this point. Following the suggestions below, we ran an analysis similar to that described in Bonaiuto et al., 2018, where we have added progressively more noise to the data to decrease the SNR. This did indeed introduce a superficial bias for all low- and high-frequency signals. The superficial bias as a function of added noise took the form of a monotonic increase followed by a saturation at higher levels of added noise. This is most clearly visible for the low-frequency analyses in new Figure 5—figure supplement 5A, B and D. For the high-frequency signals we observed a similar pattern but with two key differences- the knee of the saturating function was at a much lower added noise level (presumably corresponding to the lower signal power in the gamma range) and secondly for a number of subjects and contrasts the inference became unstable and began to flip from superficial to deep (Figure 5—figure supplement 5C, E and F) as noise was added. If the superficial localization of gamma were purely due to low SNR we would have expected the addition of noise to have little effect (i.e. the curves would be already at saturation point) also, we would not have expected the inference to flip in the opposite direction as noise was added (implying that adding this noise obscured some meaningful gamma signal).

We have described these results as follows:

“Moreover, whereas adding progressively more white noise to the sensor level data steadily increased the superficial bias of visual alpha and sensorimotor beta until a point of saturation was reached, the change in the laminar bias of visual and sensorimotor gamma saturated at a much lower noise level and became unstable for some subjects and contrasts, flipping from a superficial to a deep bias (Figure 5—figure supplement 5). If the superficial localization of gamma were a trivial consequence of low SNR we would have expected the addition of noise to have little effect (i.e. the curves would be already at saturation point) also, we would not have expected the inference to flip in the opposite direction as noise was added (implying that adding this noise obscured some meaningful gamma signal).”

4) The new figures that show the cortical distribution of the depth bias (Figures 5 and 6) reveal cortical bias-patterns that seem to match the gyral pattern of the brain. In fact, often the depth bias even seems to reverse (superficial vs. deep) depending on the position along the gyri and sulci. This is particularly apparent for all the global effects in Figure 6. Thus, there seems to be a rather strong correlation between the average depth (or lead-field strength) of a vertex pair and its depth bias, which would not be physiologically plausible. This observation is not mentioned. You should discuss it, and comment on whether or not this is problematic.

We had noticed this as well, and thanks for raising this issue. There appears to be a deep layer bias at the gyral crowns and a superficial layer bias at the sulcal fundi. Layer depths follow the curvature of the cortical surface, so the distance between layer V and the white matter surface varies from sulci to gyri (Wagstyl et al., 2018). Therefore, our use of white matter and pial surfaces as representations of deep and superficial cortical layers might indeed account for this effect. To address this potential confound we corrected for the depth bias: we used a robust linear regression to predict the difference in the change in absolute power from baseline between pial and white matter vertex pairs from the square root of the distance from each vertex pair to the scalp (averaged over each vertex in a pial/white matter vertex pair). We then ran our main laminar analysis on the residuals, thus factoring out the effect of depth. Critically, this did not qualitatively change the laminar localization of low- and high- frequency signals, suggesting that the depth bias does not trivially cause the results we report.

We have added these results to the manuscript:

“There appears to be a relationship between cortical folding and laminar bias, as evident in the cortical distribution of the difference in the unsigned fractional change in power (pial – white matter) over the whole brain (Figures 4 and 5). […] Crucially, this analysis did not change the laminar localization of low and high frequency signals (Figure 5—figure supplement 10).”

The analysis is described in the Materials and methods section:

“The analysis controlling for the effect of the distance to the scalp used robust regression (Holland and Welsch, 1977) to fit a linear model to the difference (pial – white matter) of the unsigned fractional change in power from baseline, averaged over trials. […] The main laminar analysis was then run on the residuals of this regression.”

5) You report that the gamma superficial bias did not increase with SNR for shuffled sensor data (subsection “Sensorimotor beta and gamma originate from distinct cortical laminae”, last paragraph). In contrast, Figure 5—figure supplement 4 shows a stronger superficial bias for high number of trials for gamma and shuffled sensors (gray lines rising towards the right). Do you mean "with lower SNR"? This needs to be clarified.

We apologize for the lack of clarity here. We meant to say that the effect of increasing the number of trials used in the analysis on the superficial bias of shuffled data is weaker than that of unshuffled sensor data. We have corrected this in the Results:

“Importantly, however, regardless of the SNR, the trivial superficial bias of the shuffled sensor models was weaker than that of the unshuffled sensor models, both within the functionally defined, and the anatomically constrained ROIs (Figure 5—figure supplement 4)”.

6) As noted previously, the argument is not valid, that a deep localization of beta during the pre-response suppression and during the beta-rebound is evidence against an SNR confound (subsection “Sensorimotor beta and gamma originate from distinct cortical laminae”, fourth paragraph). Beta is relatively lower during the suppression than during the rebound, but during both intervals absolute betapower is still much larger than absolute gamma power. It is this absolute power level, which is relevant for potential SNR confounds. The argument should be rephrased or removed altogether.

We agree that non-normalized, absolute power is more important for potential SNR confounds. We have therefore removed this argument from the manuscript.

7) You performed a quantification of the cross-session reproducibility. However, it seems that only the results for one subject are reported (subsection “High SNR MEG recordings using individualized head-casts”, last paragraph). The population results should be reported as well (e.g., mean +/- std).

We have now added the mean and standard deviation of ICC values across subjects for each task epoch (RDK, instruction cue, and response) and reproducibility measure (topography, ERF, and time frequency). All of the mean values were above 0.9, with the exception of the RDK-evoked response which had a mean within-session of 0.88.